# Systematic review of predictive models of microbial water quality at freshwater recreational beaches

**Cole Heasley\***, **J. Johanna Sanchez, Jordan Tustin, Ian Young**

School of Occupational and Public Health, Ryerson University, Toronto, Ontario, Canada

\* cole.heasley@ryerson.ca

**Data Availability Statement:** All relevant data are within the paper and its S1–S9 Tables and S1 Protocol files.

**Funding:** IY and JT received funding for the project by the Public Health Agency of Canada: https://

## Abstract

Monitoring of fecal indicator bacteria at recreational waters is an important public health measure to minimize water-borne disease, however traditional culture methods for quantifying bacteria can take 18–24 hours to obtain a result. To support real-time notifications of water quality, models using environmental variables have been created to predict indicator bacteria levels on the day of sampling. We conducted a systematic review of predictive models of fecal indicator bacteria at freshwater recreational sites in temperate climates to identify and describe the existing approaches, trends, and their performance to inform beach water management policies. We conducted a comprehensive search strategy, including five databases and grey literature, screened abstracts for relevance, and extracted data using structured forms. Data were descriptively summarized. A total of 53 relevant studies were identified. Most studies (n = 44, 83%) were conducted in the United States and evaluated water quality using *E. coli* as fecal indicator bacteria (n = 46, 87%). Studies were primarily conducted in lakes (n = 40, 75%) compared to rivers (n = 13, 25%). The most commonly reported predictive model-building method was multiple linear regression (n = 37, 70%). Frequently used predictors in best-fitting models included rainfall (n = 39, 74%), turbidity (n = 31, 58%), wave height (n = 24, 45%), and wind speed and direction (n = 25, 47%, and n = 23, 43%, respectively). Of the 19 (36%) studies that measured accuracy, predictive models averaged an 81.0% accuracy, and all but one were more accurate than traditional methods. Limitations identifed by risk-of-bias assessment included not validating models (n = 21, 40%), limited reporting of whether modelling assumptions were met (n = 40, 75%), and lack of reporting on handling of missing data (n = 37, 70%). Additional research is warranted on the utility and accuracy of more advanced predictive modelling methods, such as Bayesian networks and artificial neural networks, which were investigated in comparatively fewer studies and creating risk of bias tools for non-medical predictive modelling.

## Introduction

Between 2000 and 2014, 140 outbreaks were reported in 35 states and a territory in the United States (U.S.) in untreated recreational water sources, leading to 4958 cases of waterborne

www.canada.ca/en/public-health.html. Grant
number: 2021-HQ-000017. The funders had no
role in study design, data collection and analysis,
decision to publish, or preparation of the
manuscript.

**Competing interests:** The authors have declared
that no competing interests exist.

**Abbreviations:** AUC, Area under the curve;
AUROC, Area under the receiver operator curve;
CHARMS, CHecklist for critical Appraisal and data
extraction for systematic Reviews of prediction
Modelling Studies; FIB, Fecal indicator bacteria; Fn,
Fourier transform; LASSO, Least absolute
shrinkage and selection operator; NSE, Nash-
Sutcliffe efficiency; PBIAS, Percent bias; PRISMA,
Preferred Reporting Items for Systematic Reviews
and Meta-Analyses; qPCR, quantitative polymerase
chain reaction; RMSE, Root mean squared error;
U.S., United States.

disease, with 84% of the outbreaks associated with a lake, pond, or reservoir [1]. However, when accounting for non-outbreak linked cases, underreporting, and missing state data, the estimate for total water-borne illness from recreational surface waters in the U.S. is around 90 million cases annually, costing $2.2-$3.7 billion USD in healthcare services [2]. Routine monitoring for water-borne pathogens is infeasible at recreational beaches, therefore, fecal indicator bacteria (FIB) are sampled as a marker of potential pathogen concentrations and risk of infection to bathers. There are many pathogens that are spread via recreational water use that can cause recreational water illness, including enteric viruses (e.g. norovirus, adenovirus) and bacterial and protozoal pathogens (e.g. *Campylobacter*, *Salmonella*, *Cryptosporidium*) [3, 4]. *E. coli* is often used as the indicator for the presence of these pathogens in freshwater beaches [5]. *Enterococcus* is occasionally used as an indicator in addition to or in place of *E. coli*, most commonly in marine waters [6–8]. *E. coli* is often a preferred indicator in freshwater sources due to its strong association with the risk of gastrointestinal illness in bathers [5, 9].

Decisions on whether to close or post beaches as potentially unsafe for swimming due to water quality concerns are conducted by public health officials or other beach managers. Traditionally, these decisions are based on evaluating whether FIB levels in beach waters exceed health-action threshold values. This approach has been termed the "persistence model" of beach management, because it typically relies on culture-based laboratory assessments of FIB counts which require 18–24 hours to obtain a result, leading beach managers to make water quality decisions using the previous day's measurements. More modern genetic techniques, such as qPCR, can achieve results in 3–4 hours, but are costly for beach management and laboratories to run daily [10]. Some beach managers have moved to forecasting FIB levels using predictive models. These models typically use environmental inputs such as temperature, precipitation, and turbidity to predict FIB levels at beaches on a given day, which can then be validated and assessed with the subsequent FIB lab results [11, 12]. A wide variety of predictive modelling methods have been used at recreational beaches; including multiple linear regression [13, 14], artificial neural networks [15], and Bayesian networks [16]. These models use local weather and environmental data, collected from various sources, that are associated with FIB concentrations in the water [6, 17].

Given the variety of predictive modelling approaches and applications published to-date, there is a need to identify and describe existing approaches, trends, and their accuracy to inform beach water management policies. The purpose of this systematic review was to identify and summarize modelling methods used, where they have been applied, and their performance in correctly predicting beach water quality to support management decisions (e.g., posting a beach as unsuitable for swimming due to poor water quality). The review was conducted as part of a larger study to examine environmental influences on freshwater beach quality in Canada. Therefore, we have focused the scope on models developed for freshwater, recreational sites located in a temperate climate. To our knowledge, no systematic review exists on predictive models of fecal indicator bacteria at freshwater recreational sites in temperate climates.

## Methods

### Review question and eligibility criteria

The protocol for this review was created in accordance with the Preferred Reporting Items for Systematic Reviews and Meta-Analyses (PRISMA) Protocol 2015 checklist [18]. The remainder of this review was written using the PRISMA 2020 statement [19]; a PRISMA checklist is located in S1 Table. A review protocol was developed *a priori* following Cochrane Collaboration review guidelines (see S1 Protocol) [20]. However, the protocol was not registered with

any databases. The research questions were: 1) what types of predictive models were created for predicting FIB concentrations based on environmental variables for freshwater beach management decisions? 2) which predictors were included in these models? 3) how accurate are the models in determining if recreational water quality exceeds guideline recommendations?

Our eligibility criteria followed the PECO approach: Population, Exposure, Comparison, and Outcome. Our population of interest included freshwater beaches in temperate climates that are used for recreational purposes. Therefore, we excluded models focusing on coastal and estuarial waters, and waters not used for recreation (drinking water sources). Our exposure of interest included environmental data that can be collected in real time to support beach water monitoring, such as weather parameters and water conditions. We included models that compared accuracy to their original dataset, to persistent models, and that used other validation methods (e.g., bootstrapping). Our outcome of interest was FIB levels. Models predicting algal blooms were excluded. We included publications reporting on the development and/or evaluation of predictive models, reported in journal articles, conference proceedings, thesis and dissertations, and government reports. Reviews and commentary articles were excluded.

## Search strategy

We designed a comprehensive search strategy in collaboration with a research librarian. The following databases were used to search for relevant articles: Medline via OVID, SciTech Premium, Scopus, Web of Science, and ProQuest Dissertations and Thesis Global. The search terms used in each database are provided in S2 Table. As an example of the search terms used, the search in Scopus was:

(Escherichia coli OR enterococc* OR fecal indicator bacteria) AND (regression analysis OR predict* OR nowcast* OR forecast* OR model*) AND ("fresh water" OR recreational water OR beach* OR lake OR river) AND (weather OR monitor* OR rain* OR environmental).

All articles published until the search date, December 15, 2020, were included with no publication date restrictions. A grey literature search was also conducted and involved searching nine targeted government websites from December 10–14, 2020. A list of websites searched is available in the S3 Table. To ensure all relevant publications were captured, reference lists of relevant articles were hand-searched for additional potentially relevant articles.

## Relevance screening

Citations identified by the searches were stored in a Mendeley database (Elsevier, Amsterdam, Netherlands), deduplicated, and then uploaded into DistillerSR (Evidence Partners, Ottawa, Canada). All articles were independently screened twice by CH and JS in two levels of screening: title and abstract screening (Level 1) and full article screening (Level 2).

Level 1 screening involved the question:

Is this reference potentially relevant to our review? (Yes/No).

Level 2 involved three screening questions:

Is this article about microbial water quality? (e.g., measuring *E. coli*, *Enterococcus*).

Is this article about freshwater, recreational beaches in a temperate region?

Does this article report on a predictive model for beach water quality using environmental data? (Yes/No for all).

Beaches were defined as any site intended for primary water contact activities (e.g., swimming, wading, water sports) to capture all recreational water sites. All screening forms were created prior to any screening and pre-tested by two reviewers screening 50 articles and discussing discrepancies. Pre-testing of Level 1 screening resulted in a kappa score of 0.76, after

which the reviewers discussed their conflicts and agreed to proceed with independent reviewing after improving clarity on how to apply the eligibility criteria. Questions for level 2 were discussed prior to screening and tested on five articles by both reviewers to ensure consistent interpretation and clarity of the questions.

### Data characterization and extraction

Articles passing the screening process were obtained as full-texts and data were extracted using a pre-specified and pre-tested form. Data were extracted by CH into a form in DistillerSR, which can be found in S5 Table. The form included 20 questions that collected information such as location details of beaches, length of study, type of predictive model, variables explored in making the model, performance metrics of the model, and risk-of-bias. Data extraction results were independently validated by JS.

### Risk-of-bias assessment and data analysis

Risk of bias of each relevant article was assessed using the **CH**ecklist for critical **A**ppraisal and data extraction for systematic **R**eviews of prediction **M**odelling **S**tudies (CHARMS) [21]. We adapted the checklist from human health predictive models to environmental modelling. We considered "participants" to be beach days, and questions relating to human health were removed (e.g., details of treatments, blinding outcomes). Of 21 CHARMS questions, 10 were included in the data extraction form. Questions included sources of data, blinding predictors from outcomes, number and handling of missing data, predictor selection method, predictor transformations, and model validation methods and performance measures. Due to *a priori* knowledge that many studies collect data from government sources, predictor measurement methods were not included. CHARMS does not score studies based on bias, therefore, we did not determine an overall risk-of-bias score or rating for each study. Data from DistillerSR were downloaded in Excel (Microsoft, Redmond, United States) for analysis, which consisted of descriptive summary tabulations. Data visualizations were also created in Excel. While we report on performance metrics, we do not draw conclusions on validity nor compare models to each other. Meta-analysis was not deemed appropriate for this review given that predictive modelling approaches and performance metrics varied widely across studies.

## Results

Of 1710 unique citations identified in the search, 53 relevant studies were identified and included in the review (Fig 1). A descriptive summary of the model types, variables, and performances from each relevant study is presented in Table 1.

Studies were published from 2000 to 2021 (median of 2013). S6 Table summarizes study characteristics, including number of years of model building and publication type (Figs 2 and 3). While the maximum number of swimming seasons included in model building was 12 seasons [33], 19 (36%) of the studies used only one swimming season of data for model creation. Around half (26 studies, 49%) used two seasons or less. However, the number of seasons used in model building do not include seasons that were used solely for model validation in the 21 (40%) of studies that used temporal validation.

Five countries were represented in this study: U.S. (44 publications), Germany (4), Canada (2), New Zealand (2), and France (1). Additionally, the studies mostly focused on the Great Lakes, in particular Lake Michigan (20 studies) and Lake Erie (14) (Fig 4). Lake Ontario and Lake Superior were investigated in two studies each. No studies included Lake Huron. Overall, 40 studies (75%) modelled lakes and 13 studies (25%) modelled rivers. Fig 5 shows the frequency of the number of beaches in each study.

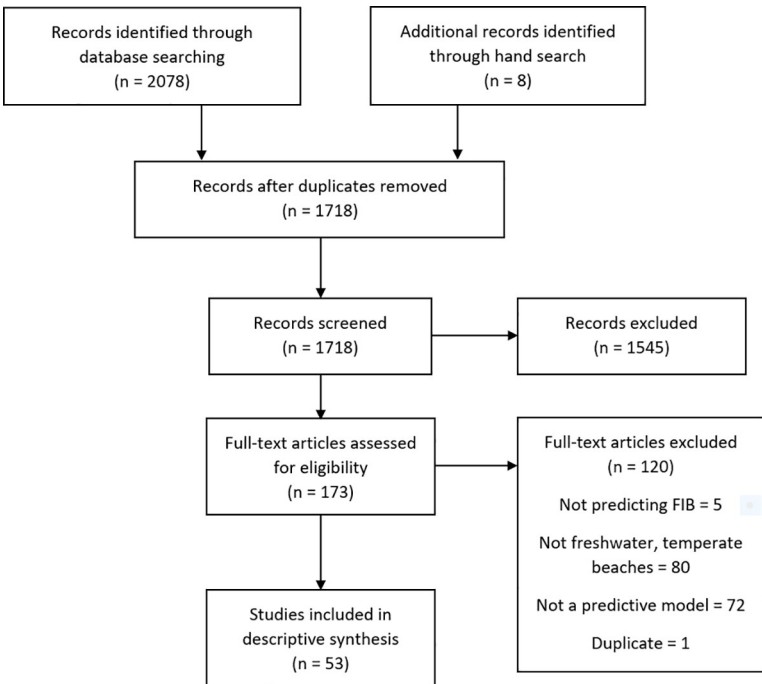

**Fig 1. PRISMA flow diagram of article selection.**

Table 2 summarizes modelling methods employed in the studies. The most commonly used model building method was multiple linear regression, which was used in 37 studies (70%), while univariate linear regression used in three (6%). Logistic regression, using a dichotomous outcome variable representing whether recreational waters met thresholds for safe use by bathers, were explored in five studies (9%). Additionally, tree regression or random forests were utilized in six studies (11%). Decision trees were created in three studies (6%). Beginning in 2012, more computationally advanced models were introduced including Bayesian networks, artificial neural networks, and deterministic or hydrodynamic models, of which there were five (9%), three (6%), and four (8%) of these model types, respectively. Various studies involved multiple modelling methods to compare their efficacy, comparing multivariate linear regression, artificial neural networks, hydrodynamic models, Bayesian networks, and stacking of multiple models together.

*E. coli* was the most commonly investigated FIB (n = 46 studies, 87%), while 11 (21%) modelled *Enterococcus*, one (2%) modelled total fecal coliforms, and one (2%) included models for *Salmonella* and *Campylobacter*. Of these, 34 (64%) studies log-transformed the concentration of the FIB of interest.

The predictor variables examined and included in final models are presented in S7 Table (and Fig 6). The variables used in most studies' final models were turbidity, wind direction, wave height, and wind speed. Time variables were important in creating models, as seen with the regular inclusion of day of year, sampling time, and month/ sub-season variables in final models. Forty-five (85%) studies assessed rainfall variables, including amount of rainfall in the previous <24, 24, 48, or 72 or more hours, the length of time since the last rainfall, or intensity of the last rainfall. Three commonly transformed variables were $\log_{10}$(turbidity), $\log_{10}$(-discharge), and weighted rainfall. Most studies obtained these environmental variables from government sources such as US Geological Survey river gauges and National Weather Service airport weather stations.

**Table 1. Summary characteristics of models extracted from 53 relevant articles that created predictive models of FIB using environmental variables.**

| | Authors and year of publication | Location of recreational waters | Number of beaches and swimming seasons [a] | Predictors explored in study | Predictors in at least one final model in study | Type of model | Model validation | Performance metrics [b] | Recommendations/ conclusions of study |
|---|---|---|---|---|---|---|---|---|---|
| 1 | Anderson, Kendall W (2019) [22] | Lake Michigan, Chicago, Illinois, USA | 19 Beaches 4 Seasons | Rainfall <24 hr, Turbidity | Rainfall <24 hr, Turbidity | Decision tree | Original dataset | Dry and wet conditions Sensitivity = 73.54% and 75.83% Specificity = 56.29% and 55.29% AUC = 69.10%, and 69.30% | Decision tree can reduce the need for qPCR testing in some conditions and issue early advisories. |
| 2 | Avila, Rodelyn, Horn, Beverley, Moriarty, Elaine, Hodson, Roger, Moltchanova, Elena (2018) [23] | Oreti river, Wallacetown, New Zealand | 1 Sampling site 9 Seasons | Rainfall 24hr, Rainfall 48 hr, Previous day [FIB], Discharge/ flow | Rainfall 24hr, Rainfall 48 hr, Previous day [FIB], Discharge/ flow | Multiple Linear Regression (MLR), Bayesian modelling, Tree regression/ random forest, Markov chain, Log-linear regression, Logistic regression, Discriminant analysis, Classification tree | Bootstrapping/ cross- validation (Leave one out and k-fold cross-validation) | Sensitivity: Dynamic regression = 62%, MLR = 62.5%, regression tree = 68%, random forest = 80%, Bayesian network = 95%, classification tree = 68%, linear discriminant analysis = 74%, Markov chain = 0%, logistic regression = 74%, quadratic discriminant analysis = 86%, random forest classification = 71% Specificity: All above 85% Error rate: dynamic MLR = 0.18, MLR = 0.21, regression tree = 0.22, random forest = 0.17, Bayesian network = 0.21, classification tree = 0.22, linear discriminant analysis = 0.19, Markov chain = 0.24, logistic regression = 0.19, Quadratic discriminant analysis = 0.26, random forest classification = 0.22 | Bayesian Networks shown to be most useful tool for prediction. |
| 3 | Bachmann-Machnik, Anna, Dittmer, Ulrich, Schoenfeld, Annika (2019) [24] | Lake Baldeney/ Ruhr River, North-Rhine-Westphalia, Germany | 1 Beach 3 Seasons | Rainfall 24hr, Sewer outflow [FIB], Discharge/ flow, Combined sewer overflow duration | No final model presented | Univariate regression | Original dataset | All $R^2 < 0.3$, none presented | Overflow events at one combined sewer outflow do not necessarily result in exceedances. |
| 4 | Brady, Amie M G, Bushon, Rebecca N, Plona, Meg B (2009) [25] | Cuyahoga River, Cuyahoga Valley National Park, Ohio, USA | 4 Sampling sites 4 Seasons | Rainfall 24hr, Turbidity, Discharge/ flow | Rainfall 24hr, Turbidity, Discharge/ flow | Multiple Linear Regression, Univariate regression | Temporal validation (new season) Geographical validation (sites3&4 modelled with site1 model) | Sensitivity: Site1 = 94%, Site2 = 100%, Sites3&4 = 73%-91% Specificity: Site1 = 64%, Site2 = 0%, Sites3&4 = 22%-96% Accuracy: Site1 = 81%, Site2 = 90%, Sites3&4 = 68%-91% | One predictive model outperformed persistence model, the other did not. Two models generalized from the first model at other locations did not outperform persistence models. |
| 5 | Brady, Amie M G, Plona, Meg B (2009) [26] | Cuyahoga River, Cuyahoga Valley National Park, Ohio, USA | 4 Sampling sites 4 Seasons | Rainfall 24hr, Turbidity, Water level | Turbidity | Univariate regression | Temporal validation (new season) | Accuracy = 77% False negative rate = 21% | Turbidity model predicted E. coli at all sites, worth posting the predicted concentrations online. |

*(Continued)*

**Table 1.** (Continued)

| | Authors and year of publication | Location of recreational waters | Number of beaches and swimming seasons [a] | Predictors explored in study | Predictors in at least one final model in study | Type of model | Model validation | Performance metrics [b] | Recommendations/ conclusions of study |
|---|---|---|---|---|---|---|---|---|---|
| 6 | Brady, Amie M.G., Plona, Meg B. (2015) [27] | Cuyahoga River, Ohio, USA | 2 Sampling sites 6 Seasons | Rainfall 24hr, Rainfall 48 hr, Water temperature, Turbidity, Discharge/flow | Rainfall 48 hr, Turbidity | Multiple Linear regression | Temporal validation (new season) | Models for most recent year. Sensitivity = 100.0% Specificity = 55.6% Accuracy = 82.2% $R^2$ = 0.77 and 0.75 RMSE = 0.3107 and 0.3197 | Automatic predictions implemented, recommend continuing nowcast system and further studies along river. |
| 7 | Brady, Amie MG, Plona, Meg B (2012) [28] | Cuyahoga River, Cuyahoga Valley National Park, Ohio, USA | 3 Sampling sites 7 Seasons | Rainfall 24hr, Rainfall 48 hr, Turbidity, Discharge/flow, Water level | Rainfall 48 hr, Turbidity, Discharge/flow, Water level | Multiple Linear Regression | Temporal validation (new season) | Sensitivity = 88% and 100% Specificity = 33% and 29% Adjusted $R^2$ = 0.69 and 0.76 RMSE = 0.34405 and 0.27396 | Predictive models performed better than persistence models in first two years, but not last year, possibly due to excess precipitation. |
| 8 | Brooks, Wesley R., Fienen, Michael N., Corsi, Steven R. (2013) [29] | Lake Erie and Lake Michigan, Cleveland and Toledo, Ohio, Port Washington, Wisconsin, USA | 4 Beaches 4 Seasons | Rainfall 24hr, Rainfall 48 hr, Air Temperature, Water temperature, Wave height, Solar radiation, Barometric pressure, Turbidity, Wind speed, Wind direction, Relative humidity, Discharge/flow, Algae index, Bird count, Lake level, Month, Day of year, Sub-season | Rainfall 24hr, Rainfall 48 hr, Air temperature, Water temperature, Wave height, Solar radiation, Barometric pressure, Turbidity, Wind speed, Wind direction, Relative humidity, Discharge/flow, Algae index, Bird count, Lake level, Month, Day of year | Multiple Linear Regression (partial least squares) | Temporal validation (new season) | Sensitivity and specificity used but values not listed. | Partial least squares automates the model building process and compares favorably to the other regression models. |
| 9 | Brooks, Wesley, Corsi, Steven, Fienen, Michael, Carvin, Rebecca (2016) [30] | Chequamegon Bay, Lake Superior and Lake Michigan, Manitowoc County, Wisconsin, USA | 7 Beaches 4 Seasons | Rainfall <24 hr, Rainfall 24hr, Rainfall 48 hr, Rainfall 72+ hr, Air Temperature, Water temperature, Wave height, Turbidity, Wind speed, Wind direction, Discharge/flow, Conductivity, Current speed, Current direction, Wave direction, Cloud cover, Bird count, Bather count, Algae presence, Day of year | Rainfall <24 hr, Rainfall 24hr, Rainfall 48 hr, Rainfall 72+ hr, Air Temperature, Water temperature, Wave height, Turbidity, Wind speed, Wind direction, Discharge/flow, Conductivity, Current speed, Current direction, Wave direction, Cloud cover, Bird count, Bather count, Algae accumulation, Day of year | Multiple Linear Regression (partial least squares and sparse partial least squares), Tree regression/ random forest, Logistic regression, Adaptive LASSO, Gradient boosting | Bootstrapping/ cross validation | AUC (AUROC): Gradient boosting cross-validation tree estimate = 0.76, Gradient boosting out-of-bag tree estimate = 0.75, MLR Adaptive LASSO = 0.73 and 0.72, Logistic regression adaptive LASSO = 0.68, 0.65, 0.63, and 0.62, Sparse partial least squares = 0.70 and 0.70, Partial Least Squares = 0.66, MLR genetic algorithm = 0.65, Logistic regression genetic algorithm = 0.60 and 0.58 | Of 14 regression methods, a random forest model was the most accurate. |
| 10 | Corsi, Steven R, Borchardt, Mark A, Carvin, Rebecca B, Burch, Tucker R, Spencer, Susan K, Lutz, Michelle A, McDermott, Colleen M, Busse, Kimberly M, Kleinheinz, Gregory T, Feng, Xiaoping, Zhu, Jun (2016) [31] | Lake Michigan, Wisconsin, USA | 3 Beaches 1 Season | Rainfall >24 hr, Rainfall 24hr, Rainfall 72+ hr, Air Temperature, Water temperature, Wave height, Turbidity, Discharge/flow, Conductivity, Water level, Current speed, Current direction, Cloud cover, Algae abundance | NA | NA | NA | Fecal indicator bacteria model not presented | NA |

*(Continued)*

**Table 1.** (Continued)

| | Authors and year of publication | Location of recreational waters | Number of beaches and swimming seasons [a] | Predictors explored in study | Predictors in at least one final model in study | Type of model | Model validation | Performance metrics [b] | Recommendations/ conclusions of study |
|---|---|---|---|---|---|---|---|---|---|
| 11 | Cyterski, M, Zhang, S, White, E, Molina, M, Wolfe, K, Parmar, R, Zepp, R (2012) [32] | Lake Michigan, Milwaukee, Wisconsin, USA | 1 Beach 1 Season | Air Temperature, Water temperature, Turbidity, Wind speed, Wind direction, Relative humidity, Conductivity, pH, Water level, Chloride, [NH₄⁺], [NO₃⁻] | Air Temperature, Water temperature, Turbidity, Wind speed, Wind direction, Relative humidity, Conductivity, pH, [NH₄⁺], [NO₃⁻], Water level | Multiple Linear Regression | Bootstrapping/ cross validation | Mean square error of prediction = 1.85, 3.67, and 2.71 | Temporal synchronization analysis of environmental predictors improved the predictive regression models. |
| 12 | Dada, Ayokunle Christopher, Hamilton, David P (2016) [33] | Lake Rotorua, North Island, New Zealand | 3 Beaches 12 Seasons | Rainfall 72+ hr, Barometric pressure, Wind speed, Wind direction, Discharge/ flow, Total nitrogen, Total phosphorus, Distance from lake exit, Suspended solids, Particulate inorganic phosphorus | Rainfall 72+ hr, Wind speed, Distance from Lake exit, Particulate inorganic phosphorus | Multiple Linear Regression | Temporal validation (2 new seasons) | Sensitivity = 0%-50% Specificity = 96.67%-100% Adjusted $R^2$ = 0.73 Accuracy = 96.67%-100% RMSE = 0.23–0.64 | Models worked well, could be used for guiding swimming advisories. |
| 13 | Francy, Donna S., Gifford, Amie M., and Darner, Robert A. (2003) [34] | Lake Erie and Mosquito Lake, Cleveland, Huntington Reservation, Lake County, and Mosquito Lake State Park, Ohio, USA | 6 Beaches 3 Seasons | Rainfall 24hr, Rainfall 48 hr, Rainfall 72+ hr, Water temperature, Wave height, Previous day [FIB], Solar radiation, Turbidity, Wind speed, Wind direction, Discharge/ flow, Conductivity, Bird count, Day of year, Water level, Current direction, Days since last rainfall | Rainfall 24hr, Rainfall 72+ hr, Wave height, Previous day [FIB], Turbidity, Wind direction, Discharge/ flow, Bird count, Current direction, Day of year, Days since last rainfall | Multiple Linear Regression | Original Dataset | $R^2$ = 0.17–0.58 Accuracy = 71.2%-90.9% False positive rate = 4.0%-15.1% False negative rate = 3.9%-14.6% | Models were beach specific, future research could test created models in future years, and test whether adding subsequent years' data improves the models. |
| 14 | Francy, D.S, Brady, A.M.G., Carvin, R.B., Corsi, S.R., Fuller, L. M., Harrison, J.H., Hayhurst, B.A., Lant, J., Nevers, M.B., Terrio, P.J., Zimmerman, T.M. (2013) [35] | Lake Michigan, Lake Erie, Lake Ontario, and Lake Superior, Illinois, Indiana, Michigan, New York, Ohio, Pennsylvania, and Wisconsin, USA | 49 Beaches 2 Seasons | Rainfall 24hr, Rainfall 48 hr, Rainfall 72+ hr, Air Temperature, Water temperature, Wave height, Solar radiation, Barometric pressure, Turbidity, Wind speed, Wind direction, Relative humidity, Discharge/ flow, Conductivity, pH, Chlorophyll a, Day of year, Bird count, Debris assessment, Dissolved O₂, Wave period, Sub-season, Current direction, Cloud cover, Water level, Algae category, Bather count, Weather category | Rainfall 24hr, Rainfall 48 hr, Rainfall 72+ hr, Air temperature, Water temperature, Wave height, Solar radiation, Barometric pressure, Turbidity, Wind speed, Wind direction, Relative humidity, Discharge/ flow, Conductivity, Chlorophyll a, Day of year, Bird count, Debris assessment, Current direction, Cloud cover, Water level, Sub-season, Algae category, Bather count, Weather category | Multiple Linear Regression | Temporal validation (new season) | Sensitivity = 0%-100% Specificity = 40.4%-100% Accuracy = 55.8%-98.4% | 24 of the 42 models performed at least 5% more accurately than persistence models. |

(*Continued*)

Table 1. (Continued)

| Authors and year of publication | Location of recreational waters | Number of beaches and swimming seasons [a] | Predictors explored in study | Predictors in at least one final model in study | Type of model | Model validation | Performance metrics [b] | Recommendations/ conclusions of study |
|---|---|---|---|---|---|---|---|---|
| 15 Francy, Donna S, Darner, Robert A (2007) [36] | Lake Erie, Cleveland and Huntington Reservation, Ohio, USA | 3 Beaches 7 Seasons | Rainfall 24hr, Rainfall 48 hr, Water temperature, Wave height, Water level, Number of wet and dry days, Day of year | Rainfall 24hr, Rainfall 48 hr, Water temperature, Wave height, Day of year | Multiple Linear Regression | Temporal validation (new season) | Sensitivity = 53.3%, 50%, 32.6% Specificity = 87.6%, 94.6%, 94.6% $R^2$ = 0.43, 0.42, 0.32 | Additional data was added to refine models. An online nowcast system implemented. |
| 16 Francy, Donna S, Stelzer, Erin A, Duris, Joseph W, Brady, Amie M G, Harrison, John H, Johnson, Heather E, Ware, Michael W (2013) [37] | Inland lakes, Ohio, USA | 13 Beaches 2 Seasons | Rainfall 24hr, Rainfall 48 hr, Rainfall 72+ hr, Water temperature, Wave height, Solar radiation, Turbidity, Wind speed, Wind direction, Discharge/ flow, Conductivity, Bird count, Bather count, Water level, Day of year | Rainfall 48 hr, Rainfall 72+ hr, Water temperature, Wave height, Turbidity, Wind speed, Wind direction, Discharge/ flow, Bather count, Water level, Bird count, Day of year | Multiple Linear Regression | Temporal validation (new season) | Sensitivity = 0%- 62.5% Specificity = 65.2%-97.8% Accuracy = 65.4%-91.8% | Three of nine site models had better accuracy, sensitivity, and specificity than persistence models, notably at two lakes with higher swimmer densities. |
| 17 Francy, Donna S., Bertke, Erin E., Darner, Robert A. (2009) [38] | Lake Erie, Huntington Reservation and Edgewater State Park, Ohio, USA | 2 Beaches 4 Seasons | Rainfall 24hr, Rainfall 48 hr, Water temperature, Wave height, Solar radiation, Turbidity, Wave period, Water level, Days since last rain, Bather count, Day of year | Rainfall 24hr, Rainfall 48 hr, Wave height, Turbidity, Water level, Day of year | Multiple Linear Regression | Original dataset | Sensitivity = 57.1% and 31.7% Specificity = 89.1% and 76.6% Accuracy = 84.9% and 61.0% | The predictive model at one beach outperformed persistence model, but the predictive model at the other beach did not. |
| 18 Francy, Donna S., Darner, Robert A., Bertke, Erin E. (2006) [39] | Lake Erie, Lorain, Huntington Reserve, Cleveland, Ohio, USA | 5 Beaches 6 Seasons | Rainfall 24hr, Rainfall 48 hr, Rainfall 72+ hr, Water temperature, Wave height, Turbidity, Wind direction, Bird count, Water level Day of year, | Rainfall 24hr, Rainfall 48 hr, Rainfall 72+ hr, Water temperature, Wave height, Turbidity, Wind direction, Water level, Day of year | Multiple Linear Regression | Original dataset | Sensitivity: Threshold method = 59.1%-92.9%, Predicted [E. coli] = 25.9%-82.1% Specificity: Threshold method = 52.6%-94.9%, Predicted [E. coli] = 63.2%-98.9% $R^2$ = 0.35-0.44 Accuracy: Threshold = 76.6%-89.7%, Predicted [E. coli] = 74.5%-88.1% | The best model made better predictions than persistence models and predictions were made available online. |
| 19 Francy, Donna S., Darner, Robert A. (2003) [40] | Lake Erie, and Mosquito Lake, Cleveland, Huntington Reservation, Ohio, USA | 4 Beaches 3 Seasons | Rainfall 24hr, Rainfall 72+ hr, Water temperature, Wave height, Solar radiation, Turbidity, Wind speed, Wind direction, Discharge/ flow, Bird count, Day of year, Current direction | Rainfall 24hr, Rainfall 72+ hr, Wave height, Turbidity, Wind direction, Discharge/ flow, Bird count, Day of year, Current direction | Multiple Linear Regression | Original dataset | $R^2$ = 0.32-0.41 Accuracy = 71.2%-90.9% False positive rate = 4.0%-14.4% False negative rate = 4.0-14.6% | Predition error too high to accurately predict E. coli concentrations, but can be used to predict the probability of exceedances. |
| 20 Frick, W.E (2006)* [41] | Lake Erie, Huntington Beach, Cleveland, Ohio, USA | 1 Beach 1 Season | Unknown | Unknown | Multiple Linear Regression | Unknown | $R^2$ and Mallow's Cp used but not values not reported | Tested the Virtual Beach program at a beach, showing the program can be helpful for creating predictive models. |

(Continued)

**Table 1.** (Continued)

| | Authors and year of publication | Location of recreational waters | Number of beaches and swimming seasons [a] | Predictors explored in study | Predictors in at least one final model in study | Type of model | Model validation | Performance metrics [b] | Recommendations/ conclusions of study |
|---|---|---|---|---|---|---|---|---|---|
| 21 | Frick, Walter E, Ge, Zhongfu, Zepp, Richard G. (2008) [42] | Lake Erie, Huntington Beach, Cleveland, Ohio, USA | 1 Beach 1 Season | Rainfall 24hr, Rainfall 48 hr, Air Temperature, Water temperature, Wave height, Solar radiation, Turbidity, Wind speed, Wind direction, Cloud cover, Dew point, Precipitation potential, Rainfall intensity | Rainfall 24hr, Wave height, Turbidity, Wind speed, Wind direction, Cloud cover, Dew point, Rainfall intensity | Multiple Linear Regression | Original dataset | Adjusted $R^2$ = 0.457–0.610 | Dynamic model built off of small amounts of data compare to static models built with more data. |
| 22 | Hatfield, Nancy Lee Clark (2000) [43] | Lake Erie and an artificial lake, Maumee Bay State Park, Ohio, USA | 2 Beaches 1 Season | Air Temperature, Water temperature, Wave height, Turbidity, Wind speed, Wind direction, Bather count, Bird count, Current direction, Wave direction, weather category, Days since last rain | Air temperature, Water temperature, Turbidity, Wind speed, Wind direction, Bird count, Bather count, Days Since last rain | Multiple Linear Regression | Original dataset | No predictive model for inland lake found. Sensitivity = 50% Specificity = 82% $R^2$ = 0.253 | A reliable model was created for the Lake Eire beach but not for the artificial lake. |
| 23 | He, Cheng, Post, Yvonne, Dony, John, Edge, Tom, Patel, Mahesh, Rochfort, Quintin (2016) [44] | Lake Ontario, Toronto, Ontario, Canada | 1 Beach 3 Seasons | Rainfall <24 hr, Air Temperature, Water temperature, Wave height, Turbidity, Wind speed, Wind direction, Discharge/ flow, Bird count, Water level, Current speed, Current direction | Rainfall <24 hr, Turbidity, Wind speed, Wind direction, Discharge/ flow | Decision tree | Temporal validation (2 new seasons) | Accuracy = 76% and 78% | Model performed better than previously developed linear regression model and persistence model. |
| 24 | Heberger, Matthew G, Durant, John L, Oriel, Kimberly A, Kirshen, Paul H, Minardi, Lee (2008) [45] | Mystic River watershed, Boston, Massachusetts, USA | 1 Sampling site 1 Season | Rainfall <24 hr, Discharge/ flow, Time since last rainfall | Rainfall <24 hr, Discharge/ flow, Time since last rainfall | Multiple Linear Regression | Temporal validation (new season) | $R^2$ (calibration) = 0.42 PRESS (calibration) = 16.9 Accuracy: correctly predicted 4/5 exceedances and 15/16 non-exceedances | Predictive models showed good agreement with models developed for other systems, and showed model perform well with rivers. |
| 25 | Herrig, Ilona, Seis, Wolfgang, Fischer, Helmut, Regnery, Julia, Manz, Werner, Reifferscheid, Georg, Boeer, Simone (2019) [46] | Rhine and Moselle rivers, Rhineland-Palatinate, Germany | 2 Beaches 2 Seasons | Rainfall 24hr, Rainfall 72+ hr, Water temperature, Solar radiation, Turbidity, Discharge/ flow, Conductivity, pH, Chlorophyll a, Dissolved $O_2$ | Rainfall 24hr, Solar radiation, Discharge/ flow | Bayesian modelling | Temporal validation (new season) | $R^2$: Site1 = 0.73, Site2 = 0.55 | Whether microbial interactions in the river are driven by hydro-meteorological factors or trophic/biotic level factors plays an important role in modelling and outcomes variables. |

(Continued)

Table 1. (Continued)

| Authors and year of publication | Location of recreational waters | Number of beaches and swimming seasons [a] | Predictors explored in study | Predictors in at least one final model in study | Type of model | Model validation | Performance metrics [b] | Recommendations/ conclusions of study |
|---|---|---|---|---|---|---|---|---|
| 26 Hong, Yi, Soulignac, Frederic, Roguet, Adelaide, Li, Chenlu, Lemaire, Bruno J, Martins, Rodolfo Scarati, Lucas, Francoise, Vincon-Leite, Brigitte (2021) [47] | Lake Créteil, Créteil, Valde-Marne, France | 3 Sampling sites 1 Season | Rainfall >24 hr, Air Temperature, Water temperature, Sewer outflow [FIB], Solar radiation, Barometric pressure, Wind speed, Wind direction, Relative humidity, Discharge/ flow, Water level, Cloud cover | Rainfall >24 hr, Air temperature, Wind speed, Wind direction, Relative humidity, Air temperature, Cloud cover | Hydrodynamic modelling | Temporal validation (new season) | $R^2$ = 0.89 NSE coefficient $\geq$ 0.7 for water flow simulations | Accurate predictions show promise for hydrodynamic modelling in stormwater systems and lakes. |
| 27 Jones, Rachael M, Liu, Li, Dorevitch, Samuel (2013) [48] | Lake Michigan, Chicago, Illinois, USA | 3 Beaches 2 Seasons | Rainfall <24 hr, Rainfall 24hr, Rainfall 48 hr, Rainfall 72+ hr, Solar radiation, Water level, Time since last rain, Rain intensity | Rainfall 48 hr, Rainfall 72+ hr, Solar radiation, Time since last rain, Intensity of rainfall | Linear mixed effects model | Division of original dataset | Ecoli: Sensitivity = 23% and 42% Specificity = 77% and 89% Accuracy = 77% and 77% Enterococcus Sensitivity = 53% and 62% Specificity = 80% and 84% Accuracy = 72% and 76% | Predictive models performed with good accuracy but low sensitivity. |
| 28 Madani, M, Seth, R (2020) [14] | Lake St. Clair, Windsor, Ontario, Canada | 1 Beach 3 Seasons | Rainfall <24 hr, Rainfall 24hr, Rainfall 72+ hr, Air Temperature, Water temperature, Wave height, Turbidity, Wind speed, Wind direction, Cloud cover, Weather category, Bird count | Rainfall <24 hr, Rainfall 24hr, Rainfall 48 hr, Rainfall 72+ hr, Air temperature, Water temperature, Wave height, Turbidity, Wind speed, Wind direction, Weather Category, Bird count | Multiple Linear Regression | Temporal validation (new season) | Sensitivity = 30%-78% Specificity = 73%-90% $R^2$ = 0.065–0.225 AUROC = 0.70–0.79 RSME = 0.251–0.449 Accuracy = 64%-78% | The predictive model outperformed persistence models. Models built using two, three, and four years of data, with the model built using two years being marginally better. |
| 29 Maimone, Mark, Crockett, Christopher S, Cesanek, William E (2007) [49] | Schuylkill River, Philadelphia, Pennsylvania, USA | 12 Sampling sites 1 Season | Turbidity, Discharge/ flow, Time since last rainfall | Turbidity, Discharge/ flow, Time since last rainfall | Decision tree | Temporal validation (new season) | Accuracy = 66% | Early testing shows model can be accurate, and when it is inaccurate, it is overly cautious, more data will be added to algorithm when available. |
| 30 Mälzer, H.-J., aus der Beek, T, Müller, S, Gebhardt, J (2016) [50] | Lake Baldeney/ Ruhr River, Northrhine-Westphalia, Germany | 1 Beach 6 Seasons | Rainfall <24 hr, Rainfall 24hr, Air Temperature, Water temperature, Solar radiation, Turbidity, Discharge/ flow, Conductivity, pH, Total and dissolved organic carbon, Spectral adsorption coefficients at 254 and 436nm, [NH$_4$+], [NO$_2$], [NO$_3$], ortho- and total-phosphate, dissolved O$_2$, Days since last rainfall | Air temperature, Water temperature, Turbidity, pH, Spectral adsorption at 254 and 436nm, [NO$_3$], [NH$_4$+], Dissolved O$_2$, Days since last rainfall, some ANN variables not listed | Multiple Linear Regression, Artificial neural networks, Deterministic (hydrodynamic) models, Logistic models | Unknown | Sensitivity: Single regression = 53%-100%, MLR = 67%-100%, ANN = 89%-100%, Logistic = 80%-100% Specificity: Single regression = 27%-68%, MLR = 0–51%, ANN = 56–83%, Logistic = 40–51% | ANN was the most accurate model, but accuracy varied across stretches of the river. |

(Continued)

**Table 1.** (Continued)

| | Authors and year of publication | Location of recreational waters | Number of beaches and swimming seasons [a] | Predictors explored in study | Predictors in at least one final model in study | Type of model | Model validation | Performance metrics [b] | Recommendations/ conclusions of study |
|---|---|---|---|---|---|---|---|---|---|
| 31 | Marion, Jason W (2011) [51] | Inland lakes, Ohio, USA | 7 Beaches 1 Season | Water temperature, Turbidity, Carlson's Trophic Index (calculated by Chlorophyll a, Total phosphorus, Secchi depth), Phycocyanin, Dissolved O₂ | Total phosphorus, Carlson's Trophic Index (calculated by Total phosphorus or mean of 3 index measures from Phosphorus, Secchi depth, and Chlorophyll a), Phycocyanin | Logistic regression | Original dataset | AUROC: TP = 0.7050, Phycocyanin = 0.6398, TSI-TP = 0.6875, TSI-mean = 0.7203 Goodness of fit test: TP = 0.1269, Phycocyanin = 0.5050, TSI-TP = 0.3588, TSI-mean = 0.3241 | Improved sensitivity is desired to reduce false negatives, but model can be useful for real-time estimates of fecal indicators. |
| 32 | Molina, M., Cyterski, Mike, Whelan, G., Zepp, R. (2014)* [52] | A Great Lake beach, USA | 1 Beach Unknown season | Unknown | Unknown | Multiple Linear regression | Unknown | Sensitivity, specificity, R² used but values not listed | Onsite data provided better predictive accuracy than publicly available data. |
| 33 | Motamarri, Srinivas, Boccelli, Dominic L (2012) [53] | Charles River Basin, Massachusetts, USA | 1 Sampling site 2 Seasons | Rainfall 24hr, Rainfall 48 hr, Previous day [FIB], Solar radiation, Discharge/ flow, Rainfall intensity, Rime since last rainfall | Rainfall 48 hr, Rainfall 72+ hr, Discharge/ flow, Rainfall intensity, Time since last rainfall (rainfall of >0.25 inches and >0.5inches) | Multiple Linear Regression, Artificial neural networks, Learning vector quantization (LVQ) | Backward elimination, LVQ used variance gained method and determinant gain method for 2 models. Top 5 variables chosen for all models. | Sensitivity: MLR = 48%, ANN = 68%, LVQ = 86% Specificity: MLR = 95%, ANN = 92%, LVQ = 98% R² and MSE: Values not reported | ANN and LVQ performed similarly, with LQV performing better with less variables included in the model. |
| 34 | Nevers, Meredith B, Shively, Dawn A, Kleinheinz, Gregory T, McDermott, Colleen M, Schuster, William, Chomeau, Vinni, Whitman, Richard L (2009) [54] | Lake Michigan, Green Bay, Sturgeon bay, Door County, Wisconsin, USA | 24 Beaches 3 Seasons | Rainfall 48 hr, Air Temperature, Water temperature, Previous day [FIB], Barometric pressure, Wind speed, Wind direction, Bird count, Water level, Wave period, Algae accumulation | Rainfall 48 hr, Water temperature, Wave height, Previous day [FIB], Barometric pressure, Wind speed, Wind direction, Bird count, Water level, Algae accumulation | Tree regression/ random forest | Original dataset | R² (adjusted R²) = 0.318 (0.315), 0.251 (0.247), 0.195 (0.184) Mallow's Cp: 6.00, 5.4, 5.68 | Models affected by generally low *E. coli* concentrations at beaches, resulting in low signal-to-noise in model building. |
| 35 | Nevers, Meredith B, Whitman, Richard L (2005) [55] | Lake Michigan, Indiana Dunes National Park, Indiana, USA | 5 Beaches 1 Season | Rainfall <24hr, Rainfall 24hr, Air Temperature, Water temperature, Solar radiation, Barometric pressure, Turbidity, Wind speed, Wind direction, Relative humidity, Discharge/ flow, Conductivity, pH, Chlorophyll a, Dew point, Wind gust, Dissolved O₂, Water level, Wave period, Wave direction, Colour, Cloud cover, Current speed, Current direction | Rainfall <24hr, Wave height, Turbidity, Chlorophyll a, Wave period | Multiple Linear Regression | Original dataset | R²: North wind = 0.6335, South wind = 0.320, North and south = 0.465 Incorrect open/closed predictions: North = 3%, south = 2% | Predictive models more accurate than persistence models. Variation better explained at beach level models. |

*(Continued)*

**Table 1.** (Continued)

| | Authors and year of publication | Location of recreational waters | Number of beaches and swimming seasons [a] | Predictors explored in study | Predictors in at least one final model in study | Type of model | Model validation | Performance metrics [b] | Recommendations/ conclusions of study |
|---|---|---|---|---|---|---|---|---|---|
| 36 | Nevers, Meredith B, Whitman, Richard L, Frick, Walter E, Ge, Zhongfu (2007) [56] | Lake Michigan, Indiana Dunes National Lakeshore, Indiana, USA | 2 Beaches 1 Season | Rainfall 24hr, Air Temperature, Water temperature, Wave height, Turbidity, Wind speed, Discharge/flow, Conductivity, pH, Chlorophyll a, Dissolved O2, Water level, Dew point, Cloud cover, Current direction, Current speed, Spectral adsorption coefficients at 254 and 436nm, Wave period | Wave height, Barometric pressure, Turbidity, Wind speed, Wind direction, Conductivity, Wave period | Multiple Linear Regression | Original dataset | $R^2$ = 0.722 and 0.504 Adjusted $R^2$ = 0.694 and 0.477 RMSE = 0.371 and 0.419 one and zero type I errors, three and two type II errors | Predictive models had less error than persistence models. Able to model beaches with multiple outfall sources. |
| 37 | Nevers, Meredith B., Whitman, Richard L. (2008) [57] | Lake Michigan, Indiana, USA | 12 Beaches 1 Season | Rainfall <24 hr, Air Temperature, Water temperature, Wave height, Previous day [FIB], Solar radiation, Barometric pressure, Turbidity, Wind speed, Wind direction, Discharge/flow, Conductivity, pH, Chlorophyll a, Colour, Dissolved O2, Wave height, Dew point, Current direction, Cloud cover | Rainfall <24 hr, Wave height, Turbidity, Wind direction, Discharge/flow | Multiple Linear Regression | Whole lake model used a division of the original dataset, while beach specific models used whole original dataset | $R^2$: All beaches = 0.48, Beach specific = 0.34–0.57 RMSE: All beaches = 0.42, Beach specific = 0.334–0.545 | Many beaches along the same coastline may be able to be modelled as if they were one beach. |
| 38 | Olyphant, G A (2005) [58] | Lake Michigan, Indiana Dunes state park, Indiana, and Lake County, Illinois, USA | 4 Beaches 1 Season | Rainfall <24 hr, Rainfall 24hr, Air Temperature, Water temperature, Wave height, Solar radiation, Wind speed, Wind direction, Water level, Streamflow [FIB], Time sample collected | Rainfall 24hr, Air temperature, Water temperature, Wave height, Solar radiation, Wind speed, Wind direction, Water level, Streamflow [FIB], Time sample collected | Multiple Linear Regression (Ordinary and generalized-least squares) | Original dataset | $R^2$ = 0.65–0.76 Accuracy = 85% - 88% | Predictive models outperformed persistence models. Model was still 90% accurate even in extreme high or low cases. |
| 39 | Olyphant, Greg A, Whitman, Richard L (2004) [59] | Lake Michigan, Chicago, Illinois, USA | 1 Beach 1 Season | Rainfall <24 hr, Rainfall 24hr, Rainfall 48 hr, Air Temperature, Water temperature, Wave height, Solar radiation, Turbidity, Wind speed, Wind direction, Conductivity, pH, Water level, Wave frequency, Dissolved O2 | Rainfall 24hr, Water temperature, Solar radiation, Turbidity, Wind speed, Wind direction, Water level | Multiple Linear Regression | Original dataset | Sensitivity = 93% Specificity = 86% Multiple correlation coefficient, R = 0.84 | Model was accurate at predicting exceedances. Large array of instrumentation tested directly on the beach. |

(*Continued*)

**Table 1.** (Continued)

| | Authors and year of publication | Location of recreational waters | Number of beaches and swimming seasons [a] | Predictors explored in study | Predictors in at least one final model in study | Type of model | Model validation | Performance metrics [b] | Recommendations/ conclusions of study |
|---|---|---|---|---|---|---|---|---|---|
| 40 | Parkhurst, David F, Brenner, Kristen P, Dufour, Alfred P, Wymer, Larry J (2005) [60] | Lake Michigan, Indiana, and Detroit river, Michigan, USA | 5 Beaches 1 Season | Air Temperature, Water temperature, Wave height, Previous day [FIB], Wind speed, Wind direction, Sunny (Y/N), Bather count, Cloud cover, Current direction, Water level, Time sample collected | Air Temperature, Water temperature, Wave height, Previous day [FIB], Wind speed, Wind direction, Sunny (Y/N), Bather count, Cloud cover, Current direction, Water level, Time sample taken | Tree regression/ random forest | Temporal validation (new season) | [E. coli] $R^2$ = 0.138 and 0.800 log[E. coli] $R^2$ = 0.824 and 0.125 | Tree regression a useful tool for exploratory analysis. Predictive model worked poorly at predicting the raw values of E. coli but worked well at predicting magnitude or log (E.coli). |
| 41 | Rossi, Alessandra, Wolde, Bernabas T., Lee, Lee H, Wu, Meiyin (2020) [61] | Passaic and Pompton rivers, New Jersey, USA | 1 Sampling site 1 Season | Rainfall <24 hr, Rainfall 24hr, Rainfall 48 hr, Rainfall 72+ hr, Air Temperature, Water temperature, Turbidity, Conductivity, pH, Chlorophyll a, Dissolved $O_2$, [$NO_3$], Dissolved organic carbon | Rainfall 72+ hr, Conductivity, pH | Logistic regression | Bootstrapping/ cross validation | $R^2$ = 0.23 and 0.41 Misclassification rate = 0.2 and 0.09. Chi-Squared statistic = 18.66 (p = 0.0001) and 23.27 (p = 0.0007) | Model shows a probabilistic measure of exceedance likelihood. Bagging technique improves reliability of model. |
| 42 | Safaie, Ammar, Wendzel, Aaron, Ge, Zhongfu, Nevers, Meredith B, Whitman, Richard L, Corsi, Steven R, Phanikumar, Mantha S (2016) [62] | Lake Michigan, Indiana Dunes National Park, Indiana, USA | 3 Beaches 1 Season | Water temperature, Solar radiation, Turbidity, Discharge/ flow, Conductivity, Current speed, Current direction | Water temperature, Solar radiation, Turbidity, Current speed, Current direction | Multiple Linear Regression and Hydrodynamic modelling | Original dataset | $R^2$: Statistical model = 0.749 and 0.710, Mechanistic = 0.603 and 0.722 RMSE: Statistical = 0.431 and 0.464, Mechanistic = 0.601 and 0.521 PBIAS: Statistical = -3.792 and -11.553, Mechanistic: 8.094 and 7.250 NSE: Statistical = 0.554 and 0.444, Mechanistic = 0.133, 0.299 RSR: Statistical = 0.677 and 0.745, Mechanistic = 0.931 and 0.837, Fn: Statistical = 0.316 and 0.333, Mechanistic = 0.440 and 0.373 | The cooperative modeling approach of using statistical models and hydrodynamic models to improve model building of the other lead to models with good predictive power that can generate real-time forecasts. |
| 43 | Seis, W, Zamzow, M, Caradot, N, Rouault, P (2018) [63] | River Havel, Berlin, Germany | 1 Sampling site 6 Seasons | Rainfall 24hr, Rainfall 48 hr, Rainfall 72+ hr, Discharge/ flow | Rainfall 24hr, Rainfall 48 hr, Rainfall 72+ hr, Discharge/ flow | Bayesian modelling | Temporal validation (2 new seasons) | Leave-one-out cross-validation information criterion = 177, 195, 191 and assessed graphically | A methodology for an early warning system, including probabilistic alert levels were developed. The model provides solutions to the current alert system. |

*(Continued)*

**Table 1.** (Continued)

| | Authors and year of publication | Location of recreational waters | Number of beaches and swimming seasons [a] | Predictors explored in study | Predictors in at least one final model in study | Type of model | Model validation | Performance metrics [b] | Recommendations/conclusions of study |
|---|---|---|---|---|---|---|---|---|---|
| 44 | Shively, Dawn A, Nevers, Meredith B, Breitenbach, Cathy, Phanikumar, Mantha S, Przybyla-Kelly, Kasia, Spoljaric, Ashley M, Whitman, Richard L (2016) [13] | Lake Michigan, Chicago, Illinois, USA | 9 Beaches 3 Seasons | Rainfall <24hr, Rainfall 24hr, Rainfall 48 hr, Air Temperature, Water temperature, Wave height, Solar radiation, Barometric pressure, Turbidity, Wind speed, Wind direction, Relative humidity, Wave period, Water level, Day of year | Air temperature, Wave height, Solar radiation, Barometric pressure, Turbidity, Wind direction, Wind speed, Day of year | Multiple Linear Regression | Temporal validation (new season) | Sensitivity = 0%–36% Specificity = 73%-100% Adjusted R² = 0.046–0.349 Accuracy = 68%–97% | Fully automated water quality system used for input into predictive model that outperformed the persistence model. Interannual model refinement improved performance. |
| 45 | Simmer, Reid A (2016) [64] | F.W. Kent Park Lake, Oxford, Iowa, USA | 1 Beach 4 Seasons | Rainfall <24hr, Rainfall 24hr, Rainfall 48 hr, Rainfall 72+ hr, Air Temperature, Water temperature, Wave height, Solar radiation, Turbidity, Wind speed, Wind direction, Relative humidity, pH, Dissolved O₂, Wave direction, Bird count, Bather count, Concentration of goose droppings, Algae presence, Day of year | Rainfall 72+ hr, Water temperature, Wind speed, Wind direction, pH, Dissolved O₂, Wave direction, Bather count, Goose dropping concentration, Day of year | Multiple Linear Regression | Bootstrapping/cross-validation | Sensitivity: 4yr = 60.00%, 2015 = 66.67% Specificity: 4yr = 90.00%, 2015 = 96.97% Adjusted R²: 4yr = 0.53, 2015 = 0.47 Accuracy: 4yr = 84%, 2015 = 90.48% (2015 being the last year modelled in the study) | Both predictive models created were more accurate than persistence models. |
| 46 | Telech, Justin W, Brenner, Kristen P, Haugland, Rich, Sams, Elizabeth, Dufour, Alfred P, Wymer, Larry, Wade, Timothy J (2009) [65] | Lake Erie and Lake Michigan, Bay Village, Ohio, Indiana Dunes National Lakeshore and Michigan City, Indiana, and St. Joseph, Michigan | 4 Beaches 2 Seasons | Rainfall 24hr, Rainfall 48 hr, Rainfall 72+ hr, Air Temperature, Water temperature, Wave height, Turbidity, Wind speed, Wind direction, pH, Cloud cover, Bather count, Bird count, Boat count, Time sample collected | Rainfall 24hr, Rainfall 48 hr, Water temperature, Wave height, Turbidity, Wind speed, Wind direction, pH, Cloud cover, Bather count, Bird count, Boat count, Time sample collected | Multiple Linear Regression | Original dataset | R² with different E. coli enumeration methods: qPCR = 0.22, 0.57, 0.39, 0.81 Membrane filtration = 0.45, 0.86, 0.50, 0.94 | Both models did not perform well at predicting Enterococcus exceedances. |
| 47 | Uejio, Christopher K, Peters, Theodore W, Patz, Jonathan A (2012) [66] | Geneva Lake, Wisconsin, United States | 5 Beaches 8 Seasons | Rainfall 24hr, Rainfall 72+ hr, Air Temperature, Wind speed, Wind direction, Discharge/flow, Days since last rainfall, Month, Sampling time | Rainfall 24hr, Rainfall 72+ hr, Wind speed, Wind direction, Discharge/flow, Month, Sampling time, Cloud cover | Bayesian modelling | Original dataset | Sensitivity = 0%–54% Specificity = 0%–80% Accuracy = 54%-100% | Predictive models at some of the beaches had good accuracy and could support decisions. |

(Continued)

**Table 1.** (Continued)

| | Authors and year of publication | Location of recreational waters | Number of beaches and swimming seasons [a] | Predictors explored in study | Predictors in at least one final model in study | Type of model | Model validation | Performance metrics [b] | Recommendations/ conclusions of study |
|---|---|---|---|---|---|---|---|---|---|
| 48 | Wang, Leizhi, Zhu, Zhenduo, Sassoubre, Lauren, Yu, Guan, Liao, Chen, Hu, Qingfang, Wang, Yintang (2020) [67] | Lake Erie, Erie county, New York, USA | 3 Beaches 7 Seasons | Rainfall 24hr, Rainfall 48 hr, Rainfall 72+ hr, Air Temperature, Water temperature, Wave height, Barometric pressure, Turbidity, Wind speed, Wind direction, Discharge/ flow, Water level, Bird count, Algae category, Debris category, Fecal matter category, Odor (Y/N), Combined sewer overflow (Y/N), Day of year, Wave direction, Cloud cover, Current speed, current direction | Rainfall 24hr, Rainfall 48 hr, Rainfall 72+ hr, Air Temperature, Water temperature, Wave height, Barometric pressure, Turbidity, Wind speed, Wind direction, Discharge/ flow, Water level, Bird count, Algae category, Debris category, Fecal matter category, Odor (Y/N), Combined sewer overflow (Y/N), Day of year, Wave direction, Cloud cover, Current speed, Current direction | Model stacking of these outputs: Multiple Linear Regression (including Partial least squares, sparse partial least squares), Bayesian modelling, Tree regression/ random forest | Bootstrapping/ cross-validation (leave one year out cross-validation) | Accuracy = 78%, 81%, and 82.3% | A model stacking approach improved robustness of prediction power, with random forest contributing the most weight in the model. |
| 49 | Wendzel, Aaron (2014) [68] | Lake Michigan, Indiana Dunes National Lakeshore, Indiana, USA | 3 Beaches 1 Season | Solar radiation, Conductivity, Current speed, Current direction, decay rate | Solar radiation, Conductivity, Current speed, Current direction, decay rate | Hydrodynamic modelling | Original dataset | RMSE = 0.600, 0.647, and 0.809 PBIAS = 6.511, 10.489, 24.283 Fourier Norm = 0.408, 0.408, 0.660 | Model could accurately simulate FIB concentrations at beaches using unstructured grids. |
| 50 | Whitman, R L, Nevers, M B (2008) [69] | Lake Michigan, Chicago, Illinois, USA | 23 Beaches 5 Seasons | Air Temperature, Water temperature, Wave height, Solar radiation, Barometric pressure, Wind speed, Wind direction, Day of year | Wave height, Barometric pressure, Day of year | Multiple Linear Regression | Original dataset | Adjusted R² = 0.20–0.41 | Beaches geographically close to each other had correlated *E. coli* fluctuations. |
| 51 | Zhang, Juan, Qiu, Han, Li, Xiaoyu, Niu, Jie, Neyers, Meredith B, Hu, Xiaonong Phanikumar, Mantha S (2018) [15] | Lake Michigan, Indiana Dunes National Park, Indiana, USA | 3 Beaches 1 Season | Rainfall 24hr, Water temperature, Wave height, Previous day [FIB], Solar radiation, Turbidity, Wind speed, Wind direction, Discharge/ flow, Conductivity, Past [FIB] beyond one day | Rainfall 24hr, Water temperature, Wave height, Turbidity, Wind speed, Discharge/ flow, Past [FIB] beyond one day | Artificial neural networks: Nonlinear input-output (NIO), nonlinear autoregressive neural network (NAR), nonlinear autoregressive network with exogenous inputs (NARX), NAR + discrete wavelet transform (WA-NAR) | Division of original dataset | Sensitivity: NIO = 0, 1, 1 NAR = 0, 0.5 0.5 NARX1 = 0, 0.5, 1 NARX2 = 0, 0.5, 1 WA-NAR = 0.5, 0.33, 0.4 Specificity: NIO = 1, 1, 1 NAR = 1, 1, 1 NARX1 = 1, 0.99, 1 NARX2 = 1, 1, 0.99 WA-NAR = 1, 1, 1 R²: NIO = 0.53, 0.43, 0.46 NAR = 0.38, 0.34, 0.59 NARX1 = 0.80, 0.82, 0.80 NARX2 = 0.77, 0.83, 0.82 WA-NAR = 0.62, 0.57, 0.62 RMSE: NIO = 0.33, 0.53, 0.32 NAR = 0.24, 0.41, 0.45 NARX1 = 0.15, 0.31, 0.23 NARX2 = 0.26, 0.23, 0.29 WA-NAR = 0.07, 0.10, 0.11 | NARX performed the best, with WA-NAR in second but requiring no explanatory variables. All models were comparable to or outperformed other predictive models previously built the these beaches. |

(*Continued*)

**Table 1.** (Continued)

| Authors and year of publication | Location of recreational waters | Number of beaches and swimming seasons [a] | Predictors explored in study | Predictors in at least one final model in study | Type of model | Model validation | Performance metrics [b] | Recommendations/ conclusions of study |
|---|---|---|---|---|---|---|---|---|
| 52 Zimmerman, Tammy M (2008) [70] | Presque Isle Beach 2, City of Erie, Pennsylvania, USA | 1 Beach 3 Seasons | Rainfall 24hr, Rainfall 48 hr, Rainfall 72+ hr, Water temperature, Wave height, Discharge/ flow, Turbidity, Wind speed, Wind direction, Conductivity, pH, Dissolved O₂ Bird count, Current speed and direction | Wave height, Turbidity, Bird count | Multiple Linear regression | Temporal validation (new season) | Sensitivity = 50.0% Specificity = 97.4% $R^2$ = 0.66 | Predictive models outperformed persistence models, notably in the models using the previous two seasons only. |
| 53 Zimmerman, Tammy M (2006) [71] | Lake Erie, City of Erie, Pennsylvania, USA | 1 Beach 2 Seasons | Rainfall 72+ hr, Water temperature, Wave height, Turbidity, Discharge/ flow, Conductivity, pH, Bird count, Debris category, Boat count, Dissolved O₂, Current direction, Current speed | Rainfall 72+ hr, Wave height, Turbidity, Wind direction | Multiple Linear regression | Original dataset | $R^2$: 2004 = 0.54, 2005 = 0.71, Both = 0.64 Log-likelihood: 2004 = -73.33, 2005 = -143.30, Both = -224.31 | Predictive models were able to predict non-exceedances well, but performed worse at predicting exceedances. |

a If different lengths of time were used at different locations, the highest number of seasons is presented. Only seasons used in model building were included, entire seasons used for model validation are not included in this count.

b Statistics for validation of models used over calibration data when available.

* Conference proceeding, only an abstract was available.

Definitions: Area under the curve (AUC), Area under the receiver operator curve (AUROC), Root mean squared error (RMSE), Percent bias (PBIAS), Least absolute shrinkage and selection operator (LASSO), Nash-Sutcliffe efficiency (NSE), Fourier transform (Fn).

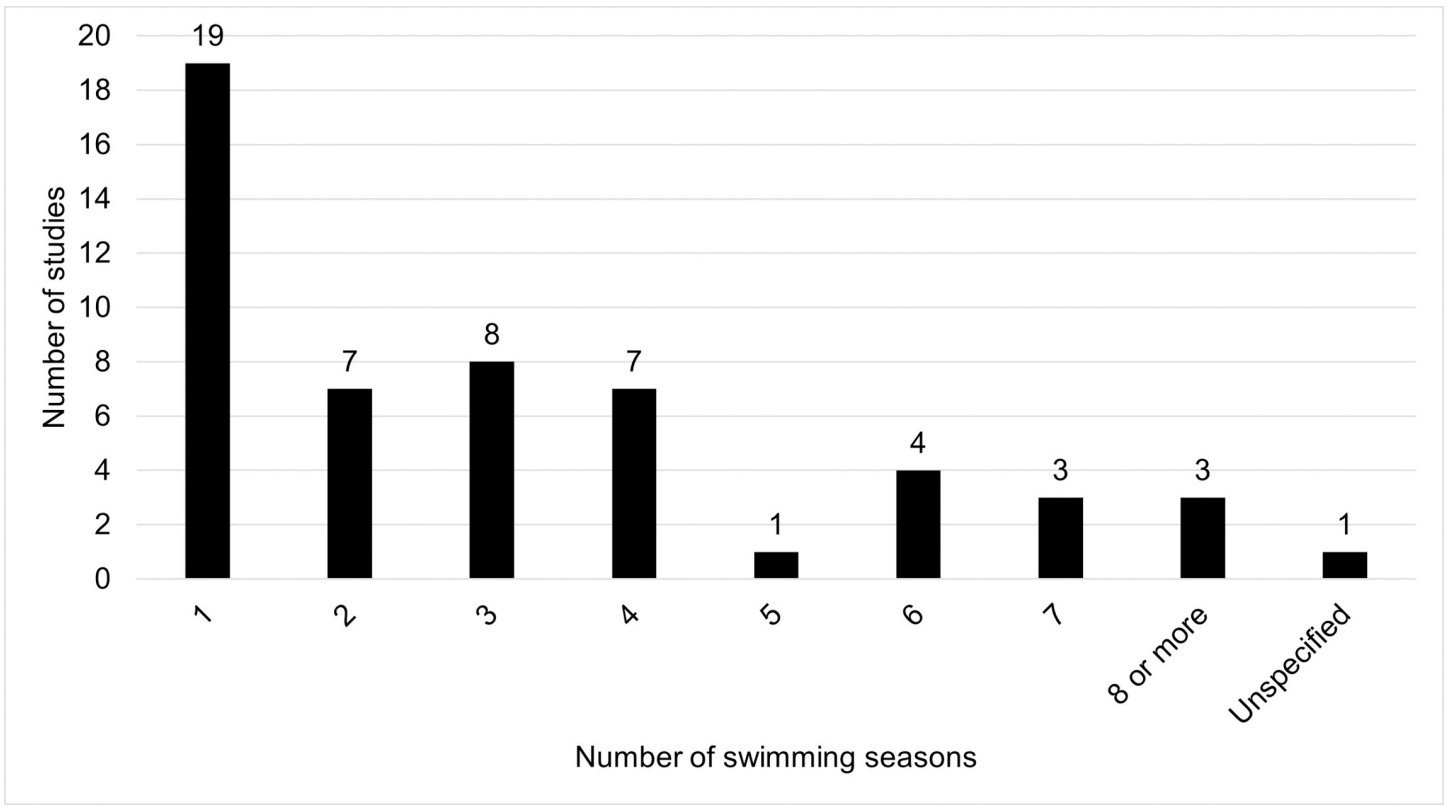

**Fig 2. Frequency of the number of swimming seasons used in building models.**

Accuracy of predictive models was measured in 19 studies. The overall accuracy of these studies was 81% (S8 Table). Of these studies, 13 compared their accuracy to pre-existing persistence models at those locations, and with the exception of one study, all or most of their models were more accurate than persistence models.

Risk-of-bias characteristics of each individual study are presented as S9 Table, while summary data are presented in Table 3. We found that one study adjusted predictor weights to address overfitting (regularization of data) and only three studies (6%) compared predictors' calibration distributions to validation distributions. Additionally, little information was provided on the handling of missing data, with only 17 (32%) studies reporting any method of dealing with missing FIB concentrations or predictor values. Modelling assumptions, such as normality, were rarely fully addressed, with only 12 (23%) studies affirming they met all model assumptions.

Predictor measurements were mostly collected from governmental sources (37 studies, 70%) or directly by the authors (28 studies, 53%) deploying their own instruments or water sampling. Most predictor transformations were categorizations (20 studies, 38%), weighting rainfall over several days (11 studies, 21%), or logarithmic (18 studies, 34%), however some studies utilized other transformations such as polynomial [64] or trigonometric transformations [34]. Twenty-seven studies (51%) reported they used no pre-screening criteria for selecting variables that were evaluated in multivariable modelling. To select predictors in final models, 13 studies (25%) used model fit characteristics of predicted values compared to actual values of FIB concentrations in many or all possible models. A full model approach using all variables was used in 10 studies (19%). Other techniques included backwards elimination,

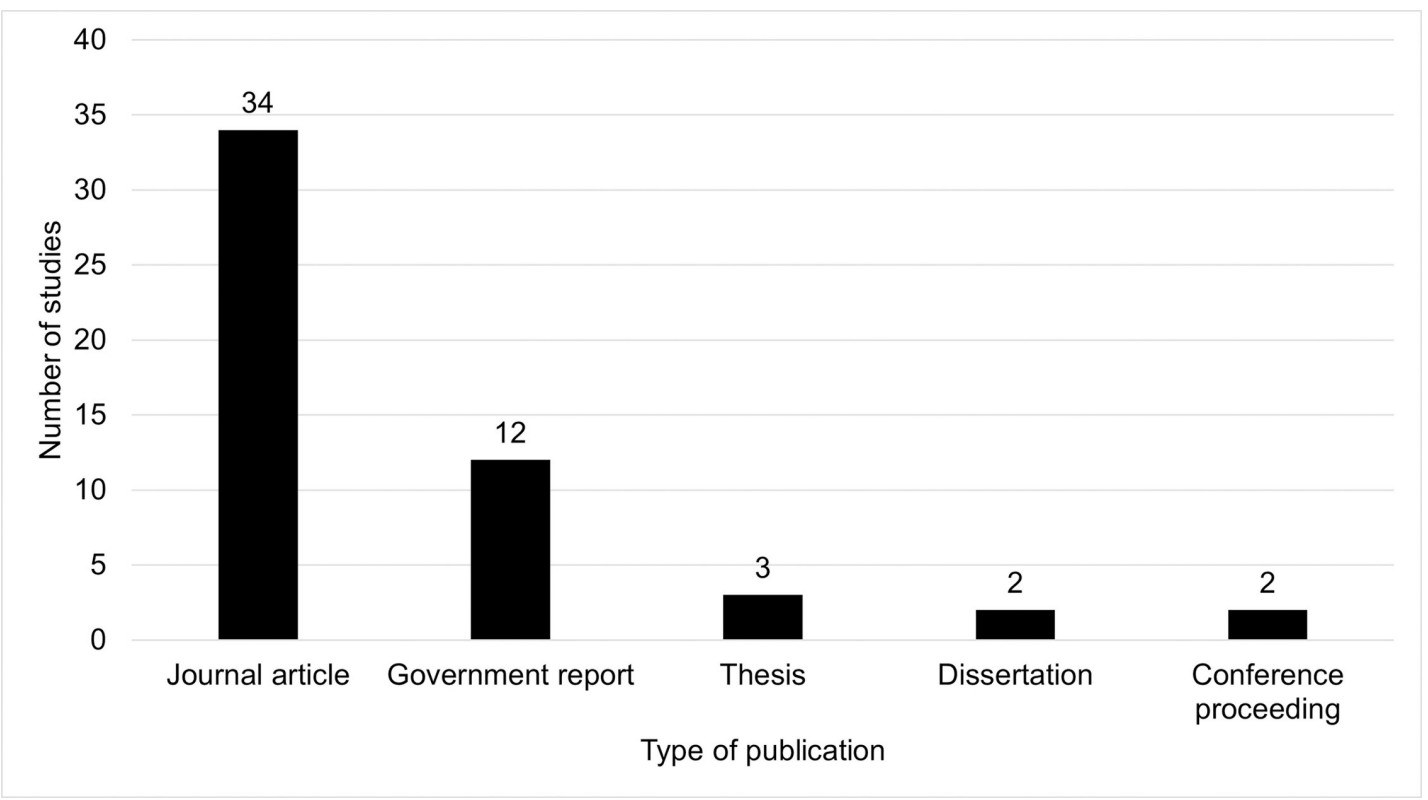

**Fig 3. Frequency of publication types.**

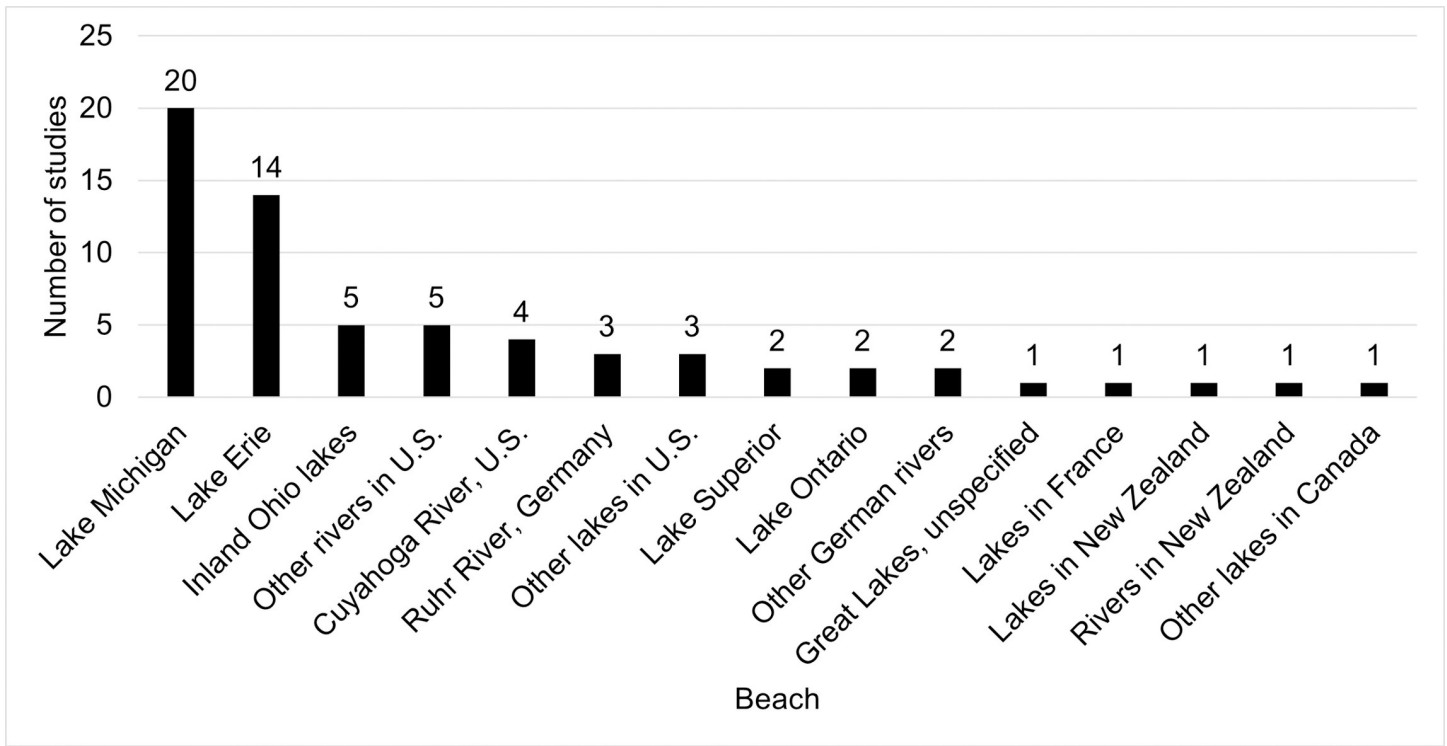

**Fig 4. Frequency of the location of beaches.**

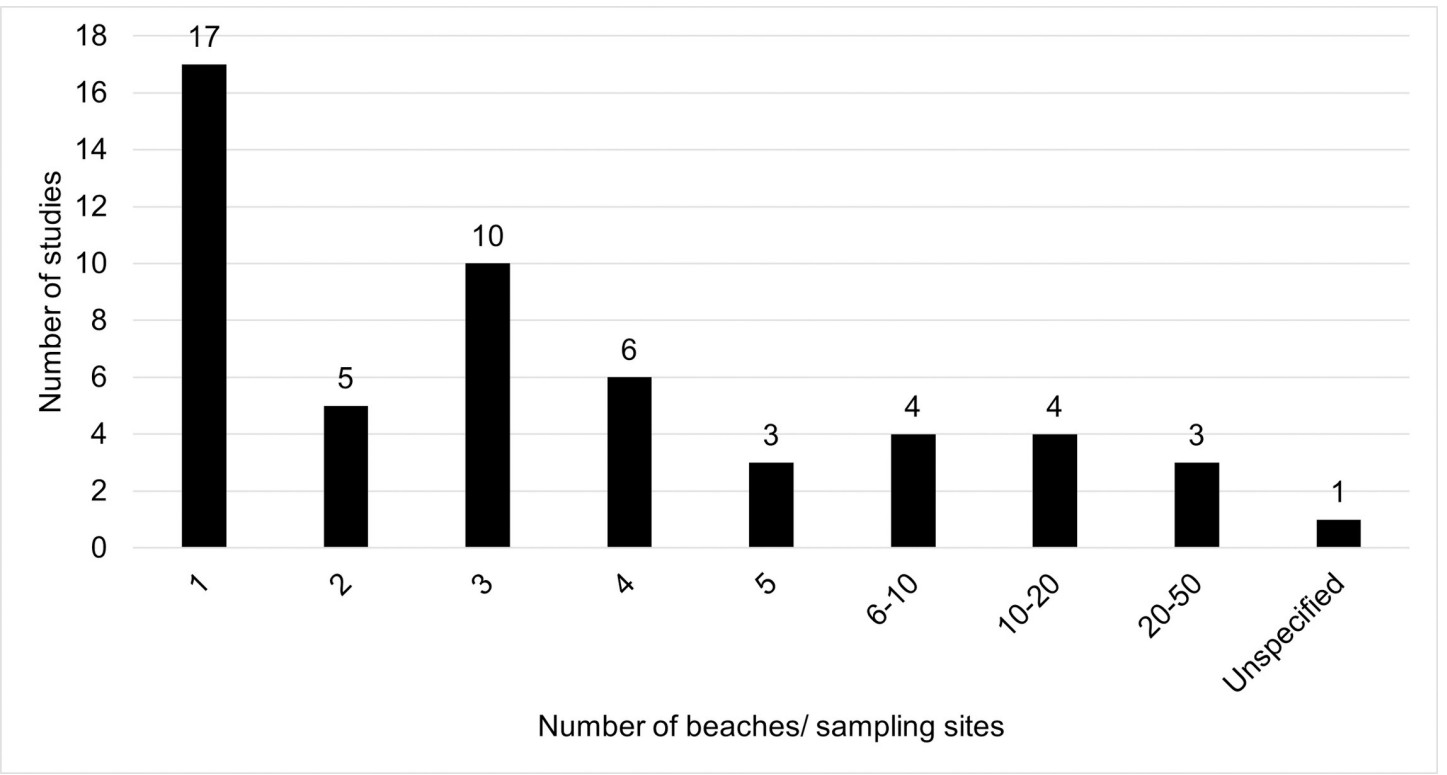

**Fig 5. Frequency of the number of beaches, or sampling sites if beaches not provided.**

Akaike's Information Criterion, and forward selection. Seven (13%) studies created models using the Virtual Beach software tool.

## Discussion

This review compiles results of the literature reporting on predictive models of FIB at fresh, recreational waters using environmental predictors. It provides novel insight on key variables of interest, modeling techniques, and considerations of modeling for those looking to create predictive models at other waters. Our review is the first to provide a systematic approach to reviewing the literature in this area. It focuses exclusively on fresh, recreational waters, and further explores the role of various environmental predictors, which is novel to the literature of

**Table 2. Modelling techniques for creating the predictive models present in 53 relevant studies.**

| Model characteristics | Number of studies | % of total studies |
|---|---|---|
| Modelling technique | | |
| Multiple linear regression | 37 | 70% |
| Tree regression and/or random forests | 6 | 11% |
| Logistic regression | 5 | 9% |
| Bayesian networks | 5 | 9% |
| Deterministic/ hydrodynamic modelling | 4 | 8% |
| Artificial neural networks | 3 | 6% |
| Univariate regression | 3 | 6% |
| Decision tree | 3 | 6% |

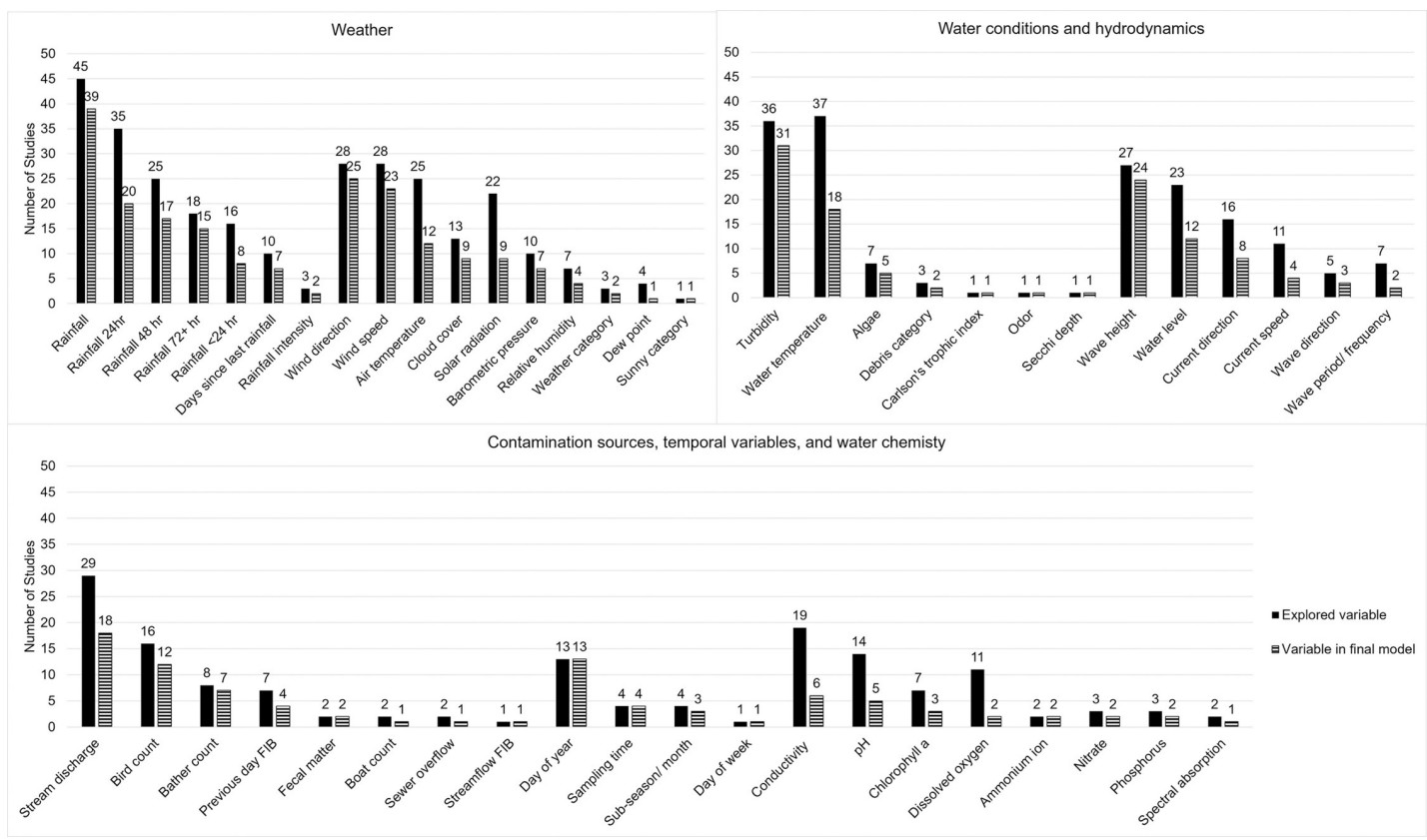

**Fig 6. Frequency of environmental variables explored in studies and frequency of variables included final models.**

this type of modelling. de Brauwere *et al.* reviewed regression and hydrodynamic models predicting FIB in all surface waters in 2014, and provided an in-depth summary of important processes for hydrodynamic models [72]. We similarly found that most relevant studies in this area were conducted in the U.S., despite wider search parameters. Additionally, this review reports on the validation techniques and amount of data used during model building and validation of reviewed studies.

As the geology, pollution sources, and climate of beaches differs geographically, building beach-specific models is important for accuracy [13, 65, 72]. Even in the same region, different bodies of water behave differently. For example, Hatfield [43] created an effective model for FIB in Lake Erie, but a similar model for a nearby artificial lake was not successful due to poor efficacy. However, geographically similar beaches within a specific region may be able to be modelled similarly to help reduce resources required to build models [54]. Different beach models may require different modeling approaches and environmental variables, so it is important to explore these elements in new contexts before generalizing models to other beaches.

Predictive modelling has the ability to overcome several issues in recreational water monitoring. Firstly, it addresses the reliance on persistence models, where the accuracy of posting beaches as suitable or unsuitable for swimming and other water activities depends on FIB concentrations remaining consistent across the 24-hour lab-response time. It also does not require the large resource and capacity investment of upgrading to qPCR for rapid testing, as most beach managers collect FIB data and government weather and water stations are already set up

**Table 3. Risk of bias checklist summary for 53 relevant studies.**

| Study design and criteria | Number of studies | % of studies |
|---|---|---|
| Source of predictor data | | |
| Government data | 37 | 70% |
| Collected by researchers | 28 | 53% |
| Collected by other researchers | 8 | 15% |
| Conservation Authorities | 3 | 6% |
| Other | 1 | 2% |
| Not clear | 6 | 11% |
| Method for selecting predictors for multivariate modeling | | |
| All included | 27 | 51% |
| Virtual Beach | 7 | 13% |
| Preselected based on significant association with FIB | 6 | 11% |
| Only univariate modeling performed | 3 | 6% |
| Preselected based on *a priori* criteria or literature | 2 | 4% |
| Other | 4 | 8% |
| Not clear | 5 | 9% |
| Predictor selection method for inclusion in final model | | |
| All possible variable combinations created, and final model chosen by model fit characteristics (e.g., $R^2$, RMSE) | 13 | 25% |
| Full model approach | 10 | 19% |
| Akaike's Information Criterion | 7 | 13% |
| Virtual Beach | 7 | 13% |
| Backward selection | 4 | 8% |
| Forward selection | 3 | 6% |
| Univariate model | 3 | 6% |
| Bayesian Information Criterion | 1 | 2% |
| Other | 7 | 13% |
| Not clear | 5 | 9% |
| Model performance measure | | |
| $R^2$ or adjusted $R^2$ | 32 | 60% |
| Sensitivity | 26 | 49% |
| Specificity | 25 | 47% |
| Accuracy | 19 | 32% |
| Root mean squared error | 8 | 15% |
| Area under the curve or area under the receiver operator curve | 4 | 8% |
| False negative or positive rate | 3 | 6% |
| Fourier transform | 2 | 4% |
| Percent bias | 2 | 4% |
| Nash-Sutcliffe efficiency | 2 | 4% |
| Mallow's Cp | 2 | 4% |
| Other | 15 | 28% |
| Model validation method | | |
| Fitting to original dataset | 21 | 40% |
| Temporal validation (new seasons) | 20 | 38% |
| Bootstrapping/ cross-validation | 6 | 11% |
| Division of original dataset | 3 | 6% |
| Geographical validation (new beaches/ sites) | 1 | 2% |
| Not clear | 2 | 4% |

*(Continued)*

**Table 3.** (Continued)

| Study design and criteria | Number of studies | % of studies |
|---|---|---|
| **Were predictor weights or regression coefficients shrunk at all?** | | |
| Yes | 1 | 2% |
| No | 52 | 98% |
| **Are modeling assumptions satisfied?** | | |
| Yes | 12 | 23% |
| No | 1 | 2% |
| Not clear | 40 | 75% |
| **Handling of predictors in modelling** | | |
| Categorized | 20 | 38% |
| Log-transformed | 18 | 34% |
| Weighted days | 11 | 21% |
| Square roots | 4 | 8% |
| Other transformations | 5 | 9% |
| **Handling of missing data** | | |
| Left as missing | 9 | 17% |
| Remove predictors with missing data | 3 | 6% |
| Data replaced with data from nearby sensor or sample collection | 3 | 6% |
| Remove days with missing predictor data | 1 | 2% |
| Autocorrelation and partial autocorrelation | 1 | 2% |
| Not clear | 37 | 70% |
| **Were predictor distributions compared between calibration and validation datasets?** | | |
| Yes | 3 | 6% |
| No | 50 | 94% |

at or near many recreational waterways, resulting in less investment to collect data to develop and implement models. However, these techniques can still be integrated together. The city of Chicago has adopted a hybrid model for determining beach water quality [73]. The five beaches (out of 20) that produce 56% of poor water quality days are tested with qPCR every-day, with the others placed into clusters, with one beach per cluster tested with qPCR and the rest predicted with models. This hybrid approach identifies poor water quality days three times more accurately than the previous predictive models alone. The rapid testing ensures accuracy, while the predictive models reduce costs and may provide a solution to the short-comings of both methods.

The efficacy of predictive models depends on the quality and accuracy of information put into them. Thirty-seven studies collected at least some of their environmental data from governmental sources, which are likely to be reliable in quality. While they might reflect slightly different weather conditions from beaches, due to being located elsewhere, such small changes are not likely to be a limitation in modelling. Rainfall is an important environmental factor as it washes microbial contamination from urban surfaces and agricultural sources into larger bodies of water, and increases sewer and river discharge [35, 47]. As a result, elevated *E. coli* levels are often associated with extreme rainfall events [69]. A wide range of timeframes for antecedent rainfall were explored, from a few hours prior to sampling to several days before. For easier interpretation, this review categorized these times as <24 hours, 24 hours, 48 hours, and 72 or more hours. Of the studies that explored times across this range, the most commonly used time in final models was 72+ hours [48, 61, 64]. Some studies also evaluated weighted

rainfall variables that emphasized more recent rainfall across a 3-day period. Regardless, when explored in a study, every rainfall variable was included in at least one final model more than 50% of the time, indicating the value of examining and comparing a variety of ways of expressing rainfall.

After rainfall, turbidity was the most frequently included variable in at least one final model. It's importance relates to the association of bacteria with sediments and particulate suspended solids [74]. As UV radiation can kill *E. coli*, higher turbidity can protect the bacteria by absorbing or scattering solar radiation [75]. The importance of sand-associated FIB was shown at a beach in Lake Huron, where erosion of sand was the main source of *E. coil* from the foreshore to surface water, mediated by wave height [76]. Larger waves may also be responsible for washing bird fecal matter from the beach into the water [54]. Wind direction and speed are important explanatory variables as they are associated with driving FIB from sediments or point sources towards the beach [77, 78]. Winds, waves, and turbidity are often correlated parameters, as winds and waves churn sediments which increases turbidity [43, 78].

While explored less often, temporal variables were consistently included in final models, 100% of the time for day of year, day of week, and time of sampling, and 75% of the time for sub-season/month. FIB may accumulate in water bodies over the summer and, on average, increase over time during the bathing season [34]. Depending on characteristics, FIB concentrations may increase as the day progresses [66] or decrease [65] due to solar inactivation. This result is also dependent on enumeration method, as Telech *et al.* found that time of day was an important predictor of *Enterococcus* cell counts, but not qPCR results [65]. Pollution sources, such as waterfowl, other bathers, and discharge into the body of water were similarly explored less often but were nonetheless important considerations.

Numerous modelling techniques and predictor selection methods were utilized in this review. Multiple linear regression methods were the most popular and were shown to produce accurate predictions. However, other methods may produce more accurate predictions. Comparing models built at different locations with different variables and rates of FIB exceedances would not yield accurate comparisons; however, four studies included in this review compared modelling techniques using the same data and were thus able to compare techniques. The best performing models in these four studies were artificial neural networks [50], Bayesian networks [23], gradient boosting machine (a type of random forest) [30], and a model stacking algorithm that combines two or more models into one prediction [67]. All outperformed regression methods such as ordinary, partial, and sparse partial least squares methods for multiple linear regression, and were more consistent across years and locations. Further research is warranted on these approaches and their utility for implementation in routine beach water quality monitoring.

Predictor selection was also varied, but no comparisons of methods were conducted. However, seven studies (13%) used the Virtual Beach tool, created by the U.S. Environmental Protection Agency, which is intended to aid researchers and beach managers in creating predictive models [79]. The tool allows users to upload data, explore relationships among variables, transform variables, use different regression-based modelling techniques (including a recent addition of a gradient boosting machine), and evaluate models based on several model fit characteristics. The tool is free and designed to be user-friendly to support implementation of modelling at more beaches. While a gradient boosting machine was added, it still relies on regression techniques. Models created by the tool outperformed persistence models in some studies [27] but not others [37].

A few key limitations in the literature were found in the risk-of-bias. For instance, 22 studies validated their models by refitting the model through the original dataset that built the model without internal validation (bootstrapping or cross-validation), which increases the risk

of overfitting [21]. Furthermore, only 13 studies (25%) specified whether or not modelling assumptions were met, which could impact model accuracy and reliability. Lastly, 37 studies (70%) did not provide any information about how missing data were dealt with, which raises additional concerns about reliability of the models. The risk of bias checklist, CHARM, required several modifications for this review compared to it's intended context of human health outcomes. A checklist intended for systematic reviews of non-health related predictive models would benefit future reviews and improve reporting of risk of bias information when creating predictive models in this research area.

The goal of predictive models is to produce more accurate results than persistence models, using the previous day's FIB measurement for current day decisions. Most models included in this review outperformed persistence models to varying degrees, in terms of sensitivity, specificity, and/or accuracy, supporting the use of predictive models in management decisions [27, 35, 64, 70, 80]. However even if models are used for management decisions, routine water sampling for FIB should still be conducted to ensure models remain valid, and are updated and refined as appropriate, across seasons. To ensure models are up to date, the U.S. Geological Survey suggests that beach managers update their predictive models before every new bathing season [27, 70], which may not always occur in practice [81].

Once an accurate model is created, their use by beach management or the public to make decisions regarding recreational activities requires a user-friendly interface. The U.S. Geological Survey Great Lakes NowCast [81] provides real-time estimates of beach water quality along Lake Erie and Lake Ontario to the public [81]. Built from the Ohio NowCast system, several studies in this review were used in developing this tool [35, 36, 38]. The predictive models created for the Cuyahoga river were also added into the Ohio NowCast [27, 28]. The website allows users to examine current and past conditions, and also explains factors in the model. The Philly Rivercast [82] provides nowcasts for the Skullykill River and it's development was outlined by Maimone *et al.* [49]. These platforms are used by beach managers and the public, which allows authorities to make real-time water quality decisions easily, and the pubic to learn about beach postings prior to arrival and make decisions about whether or not to swim or engage in other recreational activities at the beach. Additionally, as seen with the Great Lakes NowCast, these platforms can be modified and scaled to include new beaches as appropriate.

There were several limitations to this study. Firstly, while grey literature was included, only selected government websites were searched. Therefore, we could have missed some relevant studies. However, our search verification strategy helped to mitigate this potential bias. Lastly, our review was geographically limited to fresh, recreational waters in temperate regions, excluding models created for marine, tropical and subtropical waters. Predictive models in those settings may have different environmental predictors and performance.

## Conclusions

This review is the first to systematically examine literature on predictive models for FIB levels in fresh, recreational waters. The review reports on 53 relevant articles extracted from five databases. We have highlighted commonly explored and frequently used environmental variables and modelling techniques that can inform future predictive modelling projects and options for beach managers. Rainfall, turbidity, wind, and wave height were most commonly incorporated into final models, and most models used linear regression. Evidence supports use of real-time models of FIB levels as an indicator of water quality rather than or in addition to using persistence models. At locations with consistent monitoring of FIB, predictive models can improve the effectiveness and response times of risk communication with beachgoers

about recreational water quality risks, which can help to potentially reduce water-borne illness. A risk of bias checklist was adapted for this review and identified common limitations in the literature. Future research may benefit from a risk of bias checklist intended for non-medical predictive models. This review provides insight for researchers and beach managers interested in creating their own predictive models in terms of key variables, modelling approaches, and bias-reduction techniques to consider. More research should be conducted to evaluate the effectiveness and utility of more advanced predictive modelling approaches such as artificial neural networks, Bayesian approaches, and other machine learning methods.

## Supporting information

**S1 Table. PRISMA checklist for systematic reviews components and location they can be found in the review.**
(PDF)

**S2 Table. Search terms used in each database.**
(PDF)

**S3 Table. Grey literature search of government websites and their URLs.** Searched December 10–14, 2020.
(PDF)

**S4 Table. Eligibility criteria to define microbes of interest, geography, predictors of interest, and types of publications.**
(PDF)

**S5 Table. Data extraction form, including primary outcomes and risk of bias questions.**
(PDF)

**S6 Table. Descriptive summary of number of swimming seasons used for model building, number of beaches investigated, FIB of interest, geography of beaches, and type of publication of the 53 relevant studies.**
(PDF)

**S7 Table. Frequency of variables explored in studies and used in a final model for predicting microbial water quality.**
(PDF)

**S8 Table. Average accuracy of models that assessed accuracy and whether or not they performed better than persistence models.**
(PDF)

**S9 Table. Risk-of-bias characteristics of 53 articles reporting on predictive models of fecal indicator bacteria using environmental predictors, excluding characteristics found in Table 1 of main text.**
(PDF)

**S1 Protocol. Protocol for systematic literature review.**
(PDF)

## Acknowledgments

We would like to thank Cecile Farnum, a research librarian at Ryerson University, for assistance with the search strategy.

## Author Contributions

**Data curation:** Cole Heasley, J. Johanna Sanchez.

**Funding acquisition:** Jordan Tustin, Ian Young.

**Investigation:** Cole Heasley.

**Methodology:** Cole Heasley, Ian Young.

**Validation:** J. Johanna Sanchez.

**Writing – original draft:** Cole Heasley.

**Writing – review & editing:** J. Johanna Sanchez, Jordan Tustin, Ian Young.

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
