## [Decision Letter · Decision Letter 0]

22 Jul 2021

PONE-D-21-14797

Systematic Review of Predictive Models of Microbial Water Quality at Freshwater Recreational Beaches

PLOS ONE

Dear Dr. Heasley,

Thank you for submitting your manuscript to PLOS ONE. After careful consideration, we feel that it has merit but does not fully meet PLOS ONE’s publication criteria as it currently stands. Therefore, we invite you to submit a revised version of the manuscript that addresses the points raised during the review process.

We look forward to receiving your revised manuscript.

Kind regards,

Zaher Mundher Yaseen

Academic Editor

PLOS ONE

Journal Requirements:

Reviewers' comments:

Reviewer's Responses to Questions

**Comments to the Author**

1. Is the manuscript technically sound, and do the data support the conclusions?

Reviewer #1: Yes

Reviewer #2: Yes

Reviewer #3: Yes

2. Has the statistical analysis been performed appropriately and rigorously? 

Reviewer #1: Yes

Reviewer #2: Yes

Reviewer #3: Yes

3. Have the authors made all data underlying the findings in their manuscript fully available?

Reviewer #1: Yes

Reviewer #2: Yes

Reviewer #3: Yes

4. Is the manuscript presented in an intelligible fashion and written in standard English?

Reviewer #1: Yes

Reviewer #2: Yes

Reviewer #3: Yes

5. Review Comments to the Author

Reviewer #1: Interactive comment on “Systematic Review of Predictive Models of Microbial Water Quality at Freshwater Recreational Beaches” by Heasley et al.

Pr. Salim Heddam

heddamsalim@yahoo.fr.

https://orcid.org/0000-0002-8055-8463

The paper is very interesting, well written and well documented and easy to read. Few similar studies are available in the literature. In depth literature review was conducted and a useful taxonomy of predictive models of microbial water quality at freshwater (i.e. lakes and river) was provided, where standards regression and machines learning models are categorized into several groups based upon their: structure, input variables (i.e., predictor), location, number of beaches and swimming seasons, performances metrics and how the proposed models were validated against measured data, and especially, highlighting their practical benefit and environmental implications. Also, the present study provides a taxonomy that can be used to distinguish between simple, complicated, and complex models developed at different time scale. I have seen some important conclusion of the present study, which I really appreciate: (i) multiple linear regressions is used most often at nearly 70%, (ii) rainfall is reported as the most important weather variable used for model development, (iii) water turbidity (68%) and temperature (70%) are the most significant water quality variables selected as relevant predictors, (iv) in overall the proposed models were validated using R2 and the RMSE, while what surprised me mostly is that, the Nash-Sutcliffe efficiency was rarely adopted as performance metric for model validation, and (v) the most fecal indicator of interest is the E.coli. The reviewed papers were deeply analyzed and the reported results were scientifically discussed which made the manuscript very sound.

I have read the paper several times, and I see that the authors have hardly worked for providing an excellent paper and to my scientific opinion it can be accepted without revision.

Reviewer #2: The topic of this review (Systematic Review of Predictive Models of Microbial Water Quality at Freshwater Recreational Beaches) is interesting and informative for the readers. This manuscript has been written reasonably and satisfactorily, but further modifications are required before publication. My comments are:

- The abstract needs improvements and add more mathematical findings to be more informative.

- A list of abbreviations should be added.

- A list of contents should be included.

-The novelty of this work was not clearly presented. Please follow the literature review and show the knowledge gaps identified and link them to your research objectives.

- The conclusion part needs improvement to make it more informative to the readership.

Reviewer #3: Comments on the following manuscript:

Systematic Review of Predictive Models of Microbial Water Quality at Freshwater Recreational Beaches

The manuscript should be improved to visualize the results of the review conducted by the reviewers: Here my comments on the manuscript:

Abstract:

Comment No.1: Don’t use He, we in all the manuscript, revise the following sentences. And other sentences in all the manuscript. And so on.

We conducted a systematic review of predictive models of fecal

indicator bacteria at freshwater recreational sites in temperate climates to identify and describe the existing approaches, trends, and their performance to inform beach water management policies.

We conducted a comprehensive search strategy, including five databases and grey literature, screened abstracts for relevance, and extracted data using structured forms. Data

were descriptively summarized.

Comment No.2: Add recommendation in the nd of the abstract for future research.

Introduction:

Comment No.3: Check that all references cited on the manuscript.

Results:

Comment No.4: Add column in the left of Table 1 indicates the number of study from 1 to 53

Comment No.5: Add column in the right in Table 1 shows the recommendations if found of each study

Comment No.6: Add column in the center of Table 1 displays the limitations of using the proposed model of each study if found

Comment No.7: Use pie chart and column charts beside with figures to visualize your results in Table 2, 3, and 4

Especially for Table No.4, for each section, new figure can be added

Weather, hydrodynamic, contamination sources and others

Comment No.8: Use pie chart to indicate results for each contour.

Comment No.9: In your question: Were predictor weights or regression coefficients shrunk at all?

Indicate why the results 100% for No answer

Comment No.10: Add new section and talk about the accuracy and limitations of predictive models.

Comment No.11: For results in table number 5, indicate the limitation with researchers comparing with governmental for the availability of data and its effect on results.

Comment No.12: Add new table ion it compare between the used models and each limitations in prediction.

Discussion:

Provide reasons for the following conclusions:

Comment No.13: Multiple linear regression methods were the most popular and were shown to produce accurate predictions. However, other methods may produce more accurate predictions. Comparing models built at different locations with different variables and rates of FIB exceedances would not yield accurate comparisons; however, four studies included in this review compared modelling techniques.

Comment No.14: Add sentences indicate the less significant of other parameters when using the modeling rather than the parameters in the following sentences:

Larger waves may also be responsible for washing bird fecal matter from the beach into the water [54]. Wind direction and speed are important explanatory variables as they are associated with driving FIB from sediments or point sources towards the beach [77,78]. Winds, waves, and turbidity are often correlated 277 parameters, as winds and waves churn sediments which increases turbidity [43,78].

Comment No.15: Give more information about the following sentence:

Different geographical contexts require different approaches and variables, so it is important to explore these elements in new contexts.

Conclusion:

Comment No.16: Add one more recommendations for future studies in the end of conclusion section.

Comment No.17: Add symbol section definition if available to you.

6. PLOS authors have the option to publish the peer review history of their article (what does this mean?). If published, this will include your full peer review and any attached files.

Reviewer #1: **Yes: **Pr. Salim Heddam

Reviewer #2: No

Reviewer #3: No

---

## [Author Response · Author response to Decision Letter 0]

12 Aug 2021

Please see attached response to reviewers with feedback to each specific point made by the editor and reviewers.

---

## [Editor Report · Decision Letter 1]

16 Aug 2021

Systematic Review of Predictive Models of Microbial Water Quality at Freshwater Recreational Beaches

PONE-D-21-14797R1

Dear Dr. Heasley,

We’re pleased to inform you that your manuscript has been judged scientifically suitable for publication and will be formally accepted for publication once it meets all outstanding technical requirements.

Kind regards,

Zaher Mundher Yaseen

Academic Editor

PLOS ONE
---

## [Editor Report · Acceptance letter]

18 Aug 2021

PONE-D-21-14797R1 

Systematic review of predictive models of microbial water quality at freshwater recreational beaches 

Dear Dr. Heasley:

I'm pleased to inform you that your manuscript has been deemed suitable for publication in PLOS ONE. Congratulations! Your manuscript is now with our production department. 

Kind regards, 

on behalf of

Dr. Zaher Mundher Yaseen 

Academic Editor

PLOS ONE